# Harnessing Bayesian Optimism with Dual Policies in Reinforcement Learning

## Abstract

Deep reinforcement learning (RL) algorithms for continuous control tasks often struggle with a trade-off between exploration and exploitation. The exploitation objective of a RL policy is to approximate the optimal strategy that maximises the expected cumulative return based on its current beliefs of the environment. However, the same policy must also concurrently perform exploration to gather new samples which are essential for refining the underlying function approximators. Contemporary RL algorithms often entrust a single policy with both behaviours. However, these two behaviours are not always aligned; tasking a single policy with this dual mandate may lead to a suboptimal compromise, resulting in inefficient exploration or hesitant exploitation. Whilst state-of-the-art methods focus on alleviating this trade-off between exploration and exploitation to prevent catastrophic failures, they may inadvertently sacrifice the potential benefits of optimism that drives exploration. To address this challenge, we propose a new algorithm based on training two distinct policies to disentangle exploration and exploitation for continuous control and aims to strike a balance between robust exploration and exploitation. The first policy is trained to explore the environment more optimistically, maximising the upper confidence bound (UCB) of the expected return, with the uncertainty estimates for the bound derived from an approximate Bayesian framework. Concurrently, the second policy is trained for exploitation with conservative value estimates based on established value estimation techniques. We empirically verify that our proposed algorithm, combined with TD3, SAC and REDQ, significantly outperforms existing approaches across various benchmark tasks, demonstrating improved performance.

## 1 Introduction

An important phenomenon in reinforcement learning (RL) that has a complex relationship with the trade-off between exploration and exploitation (Sutton & Barto, 2018) is the overestimation bias (Thrun & Schwartz, 1993; Lan et al., 2020). State-of-the-art actor-critic RL algorithms propose to alleviate the overestimation bias to achieve better performance (Kuznetsov et al., 2020; Chen et al., 2021; Hiraoka et al., 2022), because it is often less catastrophic to underestimate the bias rather than overestimate it (Hasselt et al., 2016; Fujimoto et al., 2018). On the other hand, this bias may be viewed as a form of *optimism in the face of uncertainty*, potentially encouraging the policy to explore and take actions that may have benefits in the long run.

The trade-off between exploration and exploitation stems from demanding two distinct behaviours from a single policy. These two behaviours may often be in direct conflict; effective exploitation typically demands a near-deterministic adherence to high-value actions according to its beliefs, whereas effective exploration necessitates stochasticity and a willingness to probe seemingly suboptimal pathways. To tackle this problem, prior works have considered disentangling policies for exploration and exploitation whilst adding other techniques, such as incorporating exploration bonuses (Colas et al., 2018; Whitney et al., 2021; Schäfer et al., 2022) or using an Upper Confidence Bound (UCB) style exploration (O'Donoghue et al., 2018; Ciosek et al., 2019), or use disentangled policy trained for evaluation with off-policy RL and distribution correction (Li et al., 2022).

This work is focused on continuous control tasks. Recent works suggest that state-of-the-art RL algorithms might have been leading to too conservative value estimates, thus resulting in insufficient

exploration of the environment, especially during the initial stages of training, and suggest to amend algorithms with some optimistic value estimations (Ji et al., 2024; Omura et al., 2025). However, calculating optimistic value estimations is usually not straightforward for algorithms designed for continuous control tasks. Instead, recent works have taken inspiration from offline RL (Ji et al., 2024; Omura et al., 2025), calculates the Bellman Optimality Operator using samples from the replay buffer.

Our proposed algorithm tackles the problem from another direction. We introduce two distinct policies following the generic framework of disentangling policy learning; one trained to explore the environment optimistically and one trained to alleviate the overestimation bias by using conservative value estimates. Furthermore, instead of calculating optimistic value estimates from the replay buffer (i.e. the Bellman Optimality Operator), by disentangling the exploration and exploitation policies, we may naturally introduce optimistic state-action pairs directly into the replay buffer.

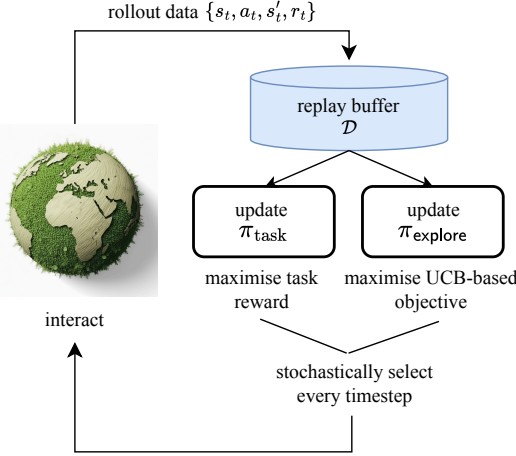

Figure 1: Schematics of our proposed method.

Our main idea is that we may utilise an approximate UCB to introduce more optimistic Q-value estimates to steer the exploration policy. We may achieve efficient exploration that cannot be achieved by simply using Gaussian noise exploration as is commonly done. Furthermore, our approach deviates from employing a weighted average of the optimistic and conservative Q-value estimates. We propose to stochastically sample from and alternate between the optimistically and conservatively trained policies during the training process. The main motivation for this design is to ensure that a capacity for exploration is retained throughout all stages of learning. This allows the policy to periodically engage in optimistic behaviours even late in the training, further preventing premature convergence to a purely exploitative policy. The schematics of our proposed method is in Figure 1.

> We term our proposed method **BOXD** (**B**ayesian **O**ptimism e**X**ploration with **D**ual Policies). The main contribution of this work is as follows.
>
> - We propose an algorithm that disentangles exploration and exploitation policies based on the Bayesian UCB principles for continuous control tasks. We utilise dropout in the Q-functions to estimate its epistemic uncertainty, and show that by calculating the maximum of Q-functions we may approximate UCB.
>
> - We propose to stochastically sample from and alternate between the optimistic and conservative trained policies during the training process, via an annealing policy conditioning scheme, to create a mixture of optimistic and conservative samples in the replay buffer.
>
> - We demonstrated that BOXD built on top of TD3, SAC and REDQ achieves considerably better performance than widely used algorithms in continuous action tasks. We argue that this outperformance stems from the usage of more optimistic exploration introduced by the disentangled exploration policy.

## 2 Preliminaries

A standard RL problem is defined as an infinite-horizon Markov Decision Process MDP = $\langle \mathcal{S}, \mathcal{A}, P, \mathcal{R}, \gamma \rangle$, where the RL agent at time $t$ observes a state $s_t$ from a set of states $\mathcal{S}$, chooses an action $a$ from a set of actions $\mathcal{A}$, and receives a reward $r$ according to a mapping of the reward function $\mathcal{R}, r : \mathcal{S} \times \mathcal{A} \to \mathbb{R}$. The environment then transitions into a state $s_{t+1}$ with a transition probability function $P(s_{t+1}|s_t, a_t)$ and the interaction continues. We also define the replay buffer $\mathcal{D}$ containing the state, action, reward, and next state at timestep t as $\mathcal{D} = (s_t, a_t, r_t, s_{t+1})$. The objective of

an RL agent is to maximise the discounted expected return $\mathbb{E}_\pi[\sum_{t=0}^\infty \gamma_t \mathcal{R}(s_t, a_t)]$, which is the expected cumulative sum of rewards when following the policy in the MDP, and the importance of the horizon is determined by a discount factor $\gamma \in [0, 1)$. Consequently, the goal is to find a policy $\pi$ that maximises the discounted expected return.

**The Bellman Equation**. In continuous RL, the Bellman equation (Richard, 1957; Sutton & Barto, 2018) play a fundamental role in defining the iterative updates for value functions in MDPs. For a given policy $\pi$, the Bellman equation describes a fundamental relationship between the value of a state-action pair $(s, a)$ and the value of the subsequent state-action pair $(s', a')$: $Q(s, a) = r + \gamma \mathbb{E}_{(s', a')}[Q(s', a')]$, where $a' \sim \pi(\cdot|s')$. In an actor-critic setting, the learning target value $y$ is set as: $y = r + \gamma Q_\phi(s', a')$, $a' \sim \pi(\cdot|s')$ and the critic objective minimisation is often calculated using Mean Squared Error (MSE) as $\mathbb{E}(y - Q_\phi(s, a))^2$.

**Bayesian Optimisation and the Upper Confidence Bound**. In Bayesian optimisation, the objective function $f(\boldsymbol{x})$ is assumed to be unknown, and the goal is to identify optimal $\boldsymbol{x}^* \in \mathcal{X}$ that maximises $f(\boldsymbol{x})$, given a set of observations $\{\boldsymbol{x}_i, y_i\}_{i=1}^N$ where $y_i = f(\boldsymbol{x}_i)$. The main challenge in Bayesian optimisation lies in effectively exploring the parameter space $\mathcal{X}$ whilst collecting informative samples. To this end, candidate points are typically selected by maximising an acquisition function $U(\boldsymbol{x})$. A widely used acquisition function is the upper confidence bound (UCB), defined as

$$U(\boldsymbol{x}) = \mu(\boldsymbol{x}) + c\sigma(\boldsymbol{x}) \tag{1}$$

where $\mu(\boldsymbol{x})$ and $\sigma(\boldsymbol{x})$ denote the predictive mean and standard deviation of $f(\boldsymbol{x})$, respectively, and $c$ is a trade-off parameter and its strength may determine the the strength of the more optimistic or conservative estimation.

In classical reinforcement learning, it has been noted that exploration using UCB often performs well in discrete state–action spaces Sutton & Barto (2018). However, in continuous state–action spaces, estimating the mean and standard deviation of the expected return is non-trivial, and therefore exploration based on the UCB principles is not readily employed in recent deep RL methods.

**Using $n$ Q-functions.** Contemporary actor-critic reinforcement learning algorithms for continuous control are predicated upon training an ensemble of $n$ independently initialised Q-functions $Q_j$ for $j = 1, 2...n$. For the computation of the target value $y$, the minimum value amongst these functions is employed: $y = r + \gamma \min_{j=1,...,n} Q_j(s', a'), a' \sim \pi(\cdot|s')$. This calculation engenders more conservative Q-value estimates (Hasselt et al., 2016; Fujimoto et al., 2018; Ciosek et al., 2019; Haarnoja et al., 2018; An et al., 2021; Chen et al., 2021; Hiraoka et al., 2022) in order to mitigate overestimation bias.

## 3 MOTIVATION: PRELIMINARY EXPERIMENTS

To provide an empirical example, we conducted an experiment in an illustrative toy environment, as depicted in the left side of Figure 2, with the goal of evaluating the convergence rate of training Q-values towards their optimal values. A reward $r_k$ is obtained upon reaching state $s_k$, whilst the episode terminates when the agent reaches either state $s_3$ or $s_4$. For this simplified scenario, the discount rate $\gamma$ is set to 0.9. Both the Q-values and the policy logits $\theta_{s,a}$ are stored in respective tabular tables, and we use two differently initialised Q-tables. The UCB is calculated as the variance between the two estimates, as shown in Equation (1). The Q values were updated using a temporal difference rule analogous to that of SARSA: $Q(s, a) \leftarrow Q(s, a) + \alpha(r + \gamma \mathbb{E}_{a' \sim \pi}[Q(s', a')])$, whilst the policy was updated using a policy gradient

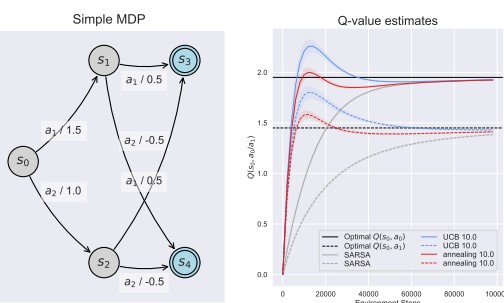

Figure 2: **Left:** A simple toy environment MDP. **Right:** The estimated values of $Q(s_0, a_0)$ and $Q(s_0, a_1)$ when using tabular actor-critic with a SARSA-based critic update, UCB-based update and annealing update. In the right sub-figure, the learning curve including the UCB may approximate the optimal value more swiftly.

method ($\theta_{s,a} \leftarrow \theta_{s,a} + \alpha \nabla_\theta \log \pi_\theta(a|s) \ Q(s,a)$), where $\alpha$ represents the step size. Some additional details about this toy example can be found in Section B.

We train two tabular Q-tables to learn the optimal Q-value for each states in the MDP in an ideal scenario (right sub-figure) without modelling noisy functional approximators. We calculate the Bayesian UCB based on the mean and the variance of two Q-tables. To obtain the best of both worlds, we also include a scheme where $Q^{\text{annealing}} = wQ^{\text{UCB}} + (1 - w)Q^{\text{SARSA}}$ that transitions from UCB to SARSA Q with $w$ decays linearly from 1.0 to 0.0.

As demonstrated in the right sub-figure in Figure 2, we may observe that SARSA-based Q updates (in grey) are less biased but potentially sacrifices convergence rate. By performing UCB-principled optimistic estimation (in red), we may obtain faster convergence to the optimal Q-value, potentially improve sampling efficiency. it is natural to use some kind of transition from optimistic UCB Q values into the less biased SARSA Q-values (in blue), as shown in the left sub-figure.

## 4 BAYESIAN OPTIMISM LEARNING WITH DUAL POLICIES

Our proposed method, **B**ayesian **O**ptimism e**X**ploration with **D**ual Policies (**BOXD**), entails training two distinct policies: an optimistic policy $\pi^{\text{explore}}$ and a conservative policy $\pi^{\text{task}}$. The optimistic policy is trained by approximating the Bayesian UCB principles. Furthermore, each policy is associated with a distinct set of Q-functions, designated as $Q^{\text{explore}}$ and $Q^{\text{task}}$ respectively. Unless specified otherwise, $Q^{\text{explore}}$ is assumed to comprise an ensemble of $n$ Q-functions, and $Q^{\text{task}}$ contains two Q-functions.

We describe how the exploration policy $\pi^{\text{explore}}$ is trained below. The conservative policy $\pi^{\text{task}}$ is trained in direct accordance with the chosen base algorithm (e.g. TD3 (Fujimoto et al., 2018), SAC (Haarnoja et al., 2018) or REDQ (Chen et al., 2021)). The pseudocode of our proposed algorithm is presented in Algorithm 1.

### 4.1 OPTIMISTICALLY TRAINED EXPLORATION POLICY

Our goal is to estimate the mean and standard deviation of the Q-function in order to calculate the UCB for a more optimistic Q-value estimation. However, in practice, directly approximating the mean and standard deviation of the Q-function is not straightforward.

Specifically, assume two samples are generated as Gaussian $Q_1, Q_2 \sim \mathcal{N}(\mu, \sigma)$. Although computing the mean and standard deviation from the generated samples is relatively simple, determining the optimism parameter in the UCB may be computationally expensive. That is, in OAC (Ciosek et al., 2019) and TOP (Moskovitz et al., 2021) the trade-off parameter $c$ in UCB is done by training/tuning additional networks. To address this, we exploit the properties of samples drawn from a Gaussian distribution. It is known that $\max(Q_1, Q_2)$ follows an extreme value distribution, and its expectation is given by

$$\mathbb{E}[\max(Q_1, Q_2)] = \mu + \frac{\sigma}{\sqrt{\pi}}, \quad (2)$$

where $\pi$ is the circle constant (Arnold et al., 1992). Leveraging this relationship, the UCB acquisition function may be approximated without the need for explicitly estimating the mean and standard deviation.

Figure 3: Best averaged IQM for 11 DM Control tasks of our proposed method versus baselines. Our proposed algorithm significantly outperforms baselines.

This relationship in equation 2 may be generalised to the maximum of $n$ samples. Each Q-function may be reformulated as $Q_j(s,a) = \mu(s,a) + \sigma(s,a)Z_j(s,a)$ where $Z_j \sim \mathcal{N}(0,1)$, the expected value of the *maximum* taken across an ensemble of $n$ Q-functions may be expressed as:

$$\mathbb{E}[\max(Q_1(s,a), Q_2(s,a), ..., Q_n(s,a))] \tag{3}$$

$$= \mathbb{E}[\max(\mu + \sigma Z_1(s,a), \mu + \sigma Z_2(s,a), ..., \mu + \sigma Z_n(s,a))] \tag{4}$$

$$= \mu + \mathbb{E}[\max(Z_1, , \ldots, Z_n)]\sigma(s,a) \tag{5}$$

$$= \mu + \int_{-\infty}^{\infty} z \cdot n[\Phi(z)]^{n-1}\phi(z)\,dz\ \sigma(s,a) \tag{6}$$

$$\approx \mu + \Phi^{-1}\left(\frac{n-0.375}{n+0.25}\right)\sigma(s,a), \qquad \text{from (Blom, 1958; Arnold et al., 1992)} \tag{7}$$

where $\Phi(z)$ denotes the standard normal cumulative distribution function (CDF) and $\phi(z)$ denotes the standard normal probability density function (PDF).

Practically, inspired by DroQ (Hiraoka et al., 2022), a recently proposed actor-critic algorithm that add dropout (Srivastava et al., 2014) into Q-functions, we model the Q-functions as stochastic functions by introducing dropout layers. We may view this use of dropout as a Bayesian approximation in Gaussian Processes (Gal & Ghahramani, 2016). Each estimate, utilising a different randomly generated dropout mask, effectively draws a sample from an approximate posterior distribution over the network's weights.

By estimating each Q-function $Q_j$ for $k$ times, both $\mu(s,a)$ and $\sigma(s,a)$ may be more stably estimated and tuned. By the derivations in Equation (3), with $a' \sim \pi^{\text{explore}}(\cdot|s')$ we therefore train our optimistic policy $\pi^{\text{explore}}$ by taking the *maximum* as:

$$Q_{\max}^{\text{explore}}(s',a') = \max(\mathbb{E}_k[Q_1^{\text{explore}}(s',a')], \mathbb{E}_k[Q_2^{\text{explore}}(s',a')], ...\mathbb{E}_k[Q_n^{\text{explore}}(s',a')]), \tag{7}$$

where $\mathbb{E}_k[Q_j^{\text{explore}}(s',a')]$ denotes the mean of the $k$ estimates sampled from the $j$-th Q-function $Q_j^{\text{explore}}$. These estimated Q-value estimates may therefore be leveraged to construct further optimistic Q-value estimates, formulated in accordance with the UCB principle, to guide the exploration policy $\pi^{\text{explore}}$. The respective target value $y$ becomes:

$$y = r(s,a) + \gamma\, Q_{\max}^{\text{explore}}(s',a') \tag{8}$$

Whilst the UCB exploration constant $c$ is typically tuned to balance exploration and exploitation in Bayesian optimisation, in our framework the degree of optimism in the Q-value estimates can instead be controlled by the number of Q-functions $n$ in the ensemble, as well as by the number of estimates $k$ generated by each $Q_j$ function. A comprehensive description of all implementation-specific details is provided in Section C.

## 4.2 Annealing Policy Conditioning When Interacting with the Environment

The interacting policy $\pi^{\text{act}}$ is chosen between $\pi^{\text{explore}}$ and $\pi^{\text{task}}$ at each timestep, instead of exclusively employing the optimistic policy $\pi^{\text{explore}}$ to interact with the environment. We propose an annealing scheme to govern this process, designed to balance the needs of exploration and exploitation during learning. This scheme begins by utilising the exploratory policy $\pi^{\text{explore}}$ at the early stages of training to ensure a rich and diverse set of samples are included in the replay buffer, before gradually transitioning to the more conservative task policy $\pi^{\text{task}}$ as learning progresses. This transition is governed by a threshold $s$, which increases linearly over the course of training in Equation (9).

Given a maximum training duration of $T$ timesteps and $\lfloor \cdot \rfloor$ denoting the floor function, our proposed annealing policy conditioning at the current timestep $t$ is expressed as:

$$s = \lfloor \frac{10t}{T} \rfloor /10, \ \text{samples}\ p \sim \mathcal{U}(0,1), \ \pi^{\text{act}} = \begin{cases} \pi^{\text{explore}} & \text{if } p > s \\ \pi^{\text{task}} & \text{otherwise} \end{cases} \tag{9}$$

---

**Algorithm 1 BOXD**

---

Initialise policy networks $\pi^{\text{explore}}$ and $\pi^{\text{task}}$, $N$ Q-function parameters $\phi_j$, $j = 1, ..., N$, and empty replay buffer $\mathcal{D}$. Set target parameters $\bar{\phi}_j \leftarrow \phi_j$, for $j = 1, ..., N$.

**while** *initial collection steps* $\leq t \leq T$ **do**

Take action $a_t \sim \pi^{\text{act}}(\cdot|s_t)$ according to annealing policy selection Equation (9). Observe reward $r_t$, next state $s_{t+1}$; $\mathcal{D} \leftarrow \mathcal{D} \bigcup (s_t, a_t, r_t, s_{t+1})$.
Sample a mini-batch $\mathcal{B} = \{(s_i, a_i, r_i, s_i')\}_{i=1}^N$ from $\mathcal{D}$.

✠   Update optimistic critic $Q_\phi^{\text{explore}}$

Compute the target value for $Q_\phi^{\text{explore}}$ by sampling $k$ times (Equation (7)):

$y_i^{\text{explore}} = r + \gamma \max_{j=1,...,N} \mathbb{E}_k[Q_{\phi_j}^{\text{explore}}(s_i', a_i')], \ a_i' \sim \pi^{\text{explore}}(\cdot|s_i')$

Update $Q_\phi^{\text{explore}}$ by minimising $N^{-1} \sum_{i=1}^N (y_i^{\text{explore}} - Q_\phi^{\text{explore}}(s_i, a_i))^2$

✠   Update conservative critic $Q_\phi^{\text{task}}$

Compute the target value for $Q_\phi^{\text{task}}$:

$y_i^{\text{task}} = r + \gamma \min_{j=1,...,N} Q_{\phi_j}^{\text{task}}(s_i', a_i'), \ a_i' \sim \pi^{\text{task}}(\cdot|s')$

Update $Q_\phi^{\text{task}}$ by minimising $N^{-1} \sum_{i=1}^N (y_i^{\text{task}} - Q_\phi^{\text{task}}(s_i, a_i))^2$

✠   Update optimistic policy $\pi^{\text{explore}}$ via base algorithm using $Q^{\text{explore}}$
✠   Update conservative policy $\pi^{\text{task}}$ via base algorithm using $Q^{\text{task}}$
✠   Update target networks if applicable, depending on the base algorithm

**end**

---

As a result, the replay buffer $\mathcal{D}$ becomes populated with a mixture of state-action pairs originating from both optimistic and conservative policies. This consolidated replay buffer is subsequently utilised for the training of both the $Q^{\text{explore}}$ and $Q^{\text{task}}$ ensembles.

This cross-collection of state-action pairs compels the conservative Q-functions to account for potentially high-reward exploratory actions, whilst simultaneously grounding the optimistic Q-functions with samples from more reliable trajectories, thereby enhancing overall learning stability. The key advantage of this methodology is that it yields less exploitative samples than would arise from interacting with the environment exclusively on samples from $\pi^{\text{task}}$, whilst producing a more tempered UCB compared to one trained solely on samples generated by $\pi^{\text{explore}}$.

## 5 EXPERIMENTS

In this section, we empirically evaluate the performance of BOXD, comparing it to previous related online RL approaches on a variety of challenging tasks. We show that BOXD outperforms previous baselines. We also provide analyses on BOXD's design choices and including ablation studies in the appendices. Detailed hyperparameters and implementation details used in our experiments are shown in Section C.

### 5.1 EXPERIMENTAL SETUPS

**Baselines.** The proposed algorithm, BOXD, is implemented upon the foundations of three widely used benchmarking algorithms for continuous control: TD3 (Fujimoto et al., 2018), SAC (Haarnoja et al., 2018) and and REDQ (Chen et al., 2021), whilst adding dropout (Srivastava et al., 2014) and layernorm (Ba et al., 2016) into Q-functions, following DroQ (Hiraoka et al., 2022). For all our algorithms, a uniform dropout rate of $p = 0.001$ is applied to all tasks.

In addition to direct comparisons with these base algorithms, we provide a comparative performance analysis against three other related methods. The first is an approach analogous to DERL (Schäfer et al., 2022), a framework which is conceptually similar in that it also disentangles exploration and exploitation policies whilst adding exploration bonuses to the exploration policy. For DERL, we

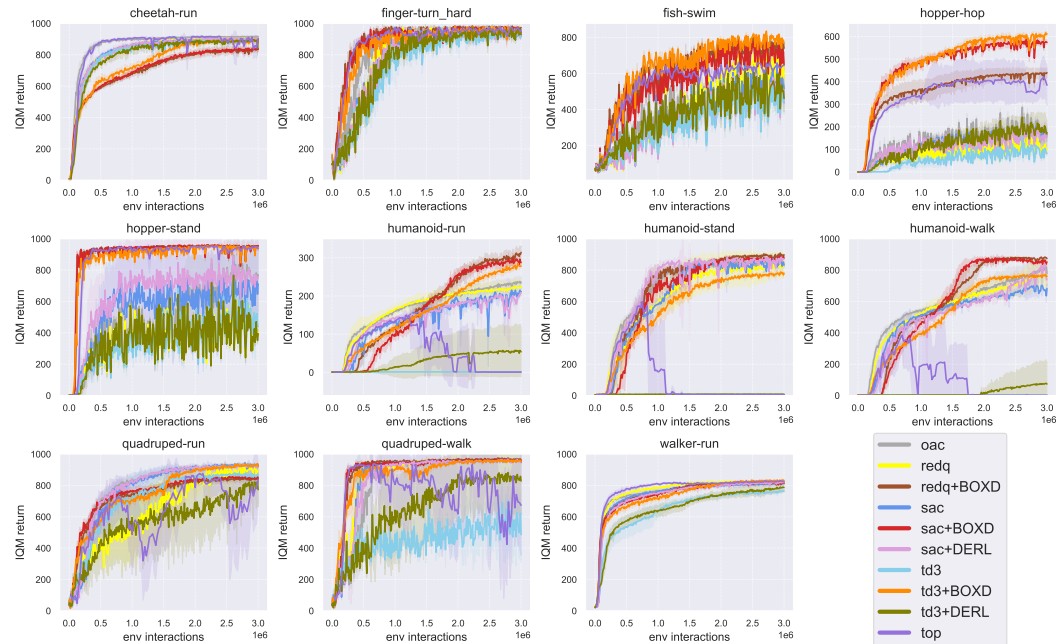

Figure 4: The IQM return for each task in DM Control of our proposed method versus baselines. Our proposed method generally achieve the best or near-best performances, whilst significantly outperforms in some tasks such as fish-swim, hopper-hop, humanoid-tasks and quadruped-walk.

also combine it with both TD3 and SAC, whilst for the exploration bonus we used RND (Burda et al., 2019). The second is OAC (Ciosek et al., 2019), a SAC-based algorithm notable for its use of optimistic Q-value estimation by UCB principles whilst addressing the directionally uninformedness of action sampled from the policy. The third one is TOP (Moskovitz et al., 2021), the source of optimism behind a more advanced state-of-the-art algorithm BRO Nauman et al. (2024), is an approach similar to OAC by learning the optimistic trade-off hyperparameter $c$ by framing it as multi-arm bandit problem.

**Benchmark and Evaluation Method.** We evaluate on 11 challenging tasks in the commonly used benchmark DeepMind Control (DM control) Suite (Tunyasuvunakool et al., 2020), where the maximum achievable return for these tasks is 1000. To show the efficacy of BOXD, we separately report 4 hard dog tasks also from DM-control and 2 sparse-reward manipulation tasks from gymnasium-robotics (de Lazcano et al., 2024). For manipulation tasks, we report the success rate. We train 10 seeds, seeds= $\{0, 1, 2, 3, 4, 5, 6, 7, 8, 9\}$, for all tasks and train for 3 million timesteps whilst evaluating every 10000 timesteps. We run 20 episodes for each evaluation, and calculate inter-quantile mean (IQM) with shaded area as the IQM-std (the std of inter-quantile samples) according to best practices (Agarwal et al., 2021). The results are shown in the next subsection.

## 5.2 RESULTS AND Q&AS

Our experiments aim to answer the following questions.

**Q: What is the performance of BOXD for DM-control benchmarking tasks ?**

**A:** Our proposed BOXD achieves the best or near-best performance on most tasks, especially in tasks where state space is larger and more exploration is desired.

The aggregated averaged IQM return across 11 tasks in DM Control is shown in Figure 3, and the IQM return for each task are shown in Figure 4. Compared with baselines, we find that BOXD generally achieves better performance or near-best with benchmarking methods. Especially in the hopper tasks and humanoid tasks, the IQM return of our proposed BOXD (paired with TD3, SAC and REDQ) significantly increases. We hypothesise that this is because these tasks have larger state dimensions, thus requiring stronger exploration mechanism in the algorithms. Whereas for cheetah-run, we hypothesise that a single mode is sufficient to achieve strong performance, therefore stronger exploration mechanism is not helpful for this task.

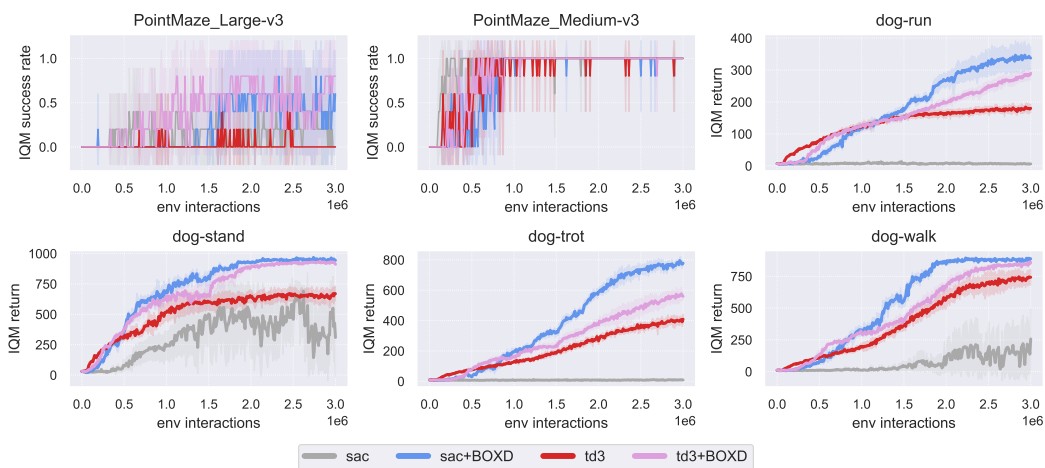

Figure 5: The IQM return for 4 hard dog tasks in DM-control and 2 sparse-reward manipulation tasks in gymnasium-robotics. Our proposed BOXD outperforms baseline algorithms by a large margin in both harder dog tasks and sparse-reward manipulation tasks.

OAC may achieve good performance on certain tasks such as cheetah-run, but underperform in some tasks such as hopper-hop. DERL paired with TD3 does not exhibit strong performance, whilst its SAC counterpart struggles at similar tasks as OAC such as hopper-hop. On the other hand, BOXD works well with both deterministic policy TD3 and stochastic policy SAC, REDQ.

An interesting phenomenon may be observed that in the humanoid-run, humanoid-stand and humanoid-walk tasks. Our proposed method exhibits a delayed performance improvement when compared to the SAC baseline. For these tasks, this initial performance lag suggests that the samples collected during the early, exploration-focused phase do not yield immediate benefits for exploitation. However, these exploration samples prove to be important in the later stages of training. Collectively, these findings underscore a key insight: enhancing the exploratory process during the initial phases of training may ultimately lead to superior asymptotic performance and improved overall sample efficiency.

**Q: What is the performance of BOXD for DM-control dog tasks and sparse-reward tasks?**
**A:** Our proposed BOXD achieves the best or near-best performance.

On these tasks, we trained our proposed BOXD using default settings ($n = 2$, $k = 2$). For dog tasks, BOXD outperforms baselines by a substantial margin, achieving state-of-the-art performance whilst being much simpler to implement compared to other state-of-the-art algorithms that incorporates complex components (Nauman et al., 2024; Lee et al., 2025). For the harder sparse-reward manipulation tasks, i.e. PointMaze-Large, our proposed BOXD can generate successful episodes whilst baseline algorithms fail to do so. The IQM return/success rate for each task are shown in Subsection 5.2.

**Q: What is the effect of annealing policy conditioning? Can we not just use $\pi^{\text{explore}}$?**
**A:** We can, and it performs well in general. However, without transitioning from the exploration policy $\pi^{\text{explore}}$ to the exploitation policy $\pi^{\text{task}}$ (i.e. annealing policy conditioning), the performance will saturate at some point.

To verify the effectiveness of our proposed annealing policy conditioning, we trained additionally on all tasks with four more patterns. Firstly, with no conditioning, where we always use $\pi^{\text{explore}}$ as acting policy $\pi^{\text{act}}$ (i.e. setting $s = 0$ as threshold, noted as 100-0); secondly, a fixed 10% probability to use $\pi^{\text{task}}$ for sampling the action policy $\pi^{\text{act}}$ (i.e. setting $s = 0.1$ as threshold, noted as 90-10); a 50% probability (i.e. setting $s = 0.5$ as threshold, noted as 50-50) to choose between $\pi^{\text{task}}$ and $\pi^{\text{act}}$; and finally a purely linear annealing conditioning with $s = t/T$. The aggregated averaged IQM return over all 11 tasks is shown in Figure 6, and individual performance for each task and be found in Section F. For these experiments, we use fixed $k = 2$ and $n = 2$ for easier comparisons.

We may observe in Figure 6 that without the annealing conditioning strategy, the performance will stagnate or decay at the later stages of the training. We hypothesise that this is because the re-

play buffer will contain fewer samples that are aligned with exploitation objective, whilst having too much exploration-aligned samples. Some form of transitioning from $\pi^{\text{explore}}$ to $\pi^{\text{task}}$ may be beneficial, also shown in the toy example Section 3. To our surprise, a strategy of 50-50 probability sampling between the two exploration and exploitation policies performs remarkably well. Nevertheless, even without the annealing conditioning strategy, our proposed method outperforms baseline algorithms.

**Q: What are the important hyperparameters of BOXD?**

**A:** The most important hyperparameter is the number of sampling times, $k$, for the Q-value estimates. In general, setting it to $k = 2$ is sufficient.

The main hyperparameter introduced by our method is the number of sampling times $k$ utilised for the computation of the optimistic Q-value estimate, $Q_j^{\text{explore}}$. Empirically, we find that a value of $k = 2$ is sufficient for obtaining a robust estimate. For the conservative counterpart, $Q_j^{\text{task}}$, a single sample is employed (i.e. $k = 1$), a configuration consistent with standard practice in related algorithms when dropout is used (e.g. DroQ (Hiraoka et al., 2022)). The second hyperparameter related to optimism is

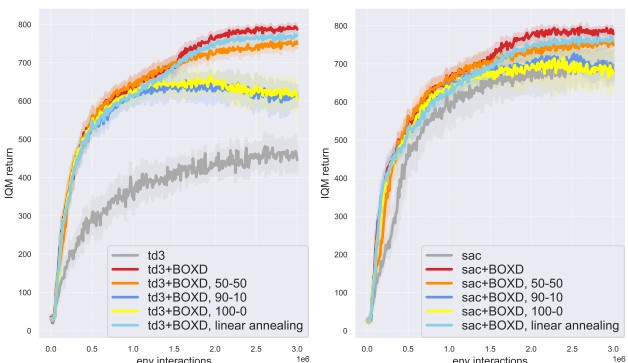

Figure 6: The average IQM for 11 DM Control tasks of our proposed method with or without annealing conditioning. **Left:** TD3-based. **Right:** SAC-based. Our proposed annealing strategy proves to be important. No conditioning or fixed conditioning show performance saturation.

the number of Q-functions $n$ used in calculating the optimistic Q-value estimates. Generally we find that setting $n = 2$ (i.e. the same as $Q^{\text{task}}$) is a good start.

We provide a comprehensive ablation studies regarding the number of critic samples $k$ and the number of Q-functions $n$. We stress that there is not much need to tune hyperparameters for each individual tasks, as the default performance already outperforms baselines. For completeness, the result using default hyperparameter values ($n = 2$, $k = 2$) are also shown. Additionally, the dropout rate can also affect performance. An ablation study of the influence of dropout rate is included. Please refer to Section D for these ablation studies.

**Q: What about more recent state-of-the-art algorithm, such as BRO?**

**A:** Comparing directly with BRO (Nauman et al., 2024) is not so straightforward because BRO employs a much larger and deeper networks, and since our proposed method mainly address how the optimism is encouraged, to ensure fairer comparison we compare with BRO's optimistic component, namely TOP (Moskovitz et al., 2021). The results are included in Figure 4. For more details, please refer to Section C. For completeness, we also provide comparison directly with BRO in Section E, showing that BOXD can achieve best or near-best performance whilst having less than half computational costs, shown in Table 1.

**Q: What are the computational costs?**

**A:** We perform computational cost calculations comparing BOXD with baseline algorithms on one of the tasks, walker-run in a P-100 GPU in Table 1. As expected, the training time of our proposed method is approximately twice of the baseline we built on, since we have two copies of networks. However, our training time is still faster compared to BRO (Nauman et al., 2024) which use much deeper networks for their critic networks. We show in Section E that BOXD can achieve near-best performance compared to BRO, whilst using approximately half of the computation cost.

Table 1: Reference computational cost on walk-run. Our proposed method can achieve similar performance as BRO in approximately half of the training time.

| algorithm | TD3 | TD3-BOXD | SAC | SAC-BOXD | TOP | BRO |
|---|---|---|---|---|---|---|
| approx. time | 11hrs | 18hrs | 12hrs | 20hrs | 13.5hrs | 36hrs |

## 6 RELATED WORKS

**Disentangle Exploration and Exploitation.** Several works have previously explored the idea of disentangling exploration and exploitation policies by adding exploration bonuses to the exploration policy DERL (Schäfer et al., 2022), GEPPG (Colas et al., 2018); by training/tuning a trade-off parameter $c$ in UCB principles (OAC (Ciosek et al., 2019)), (TOP (Moskovitz et al., 2021)); or by changing different objectives (DEEP) (Whitney et al., 2021). The work of Beyer et al. (2019) is similar to ours where they train multiple policies and choose different policies to interact with the environment. Our work mainly differ from these lines of work in the way exploration is encouraged. Our work proposes to leverage Bayesian principles by taking the maximum of dropout-enabled Q functions, which is significantly easier to implement and tune compared to the learnt optimism in DERL, OAC and TOP.

In meta-RL, (Liu et al., 2021; Norman & Clune, 2024) decouples exploration and exploitation policies where exploration and exploitation is not done concurrently rather as a prerequisite for meta-RL tasks, whilst (Liu et al., 2021) constructed separated exploitation objective from exploration, whilst automatically identify and recover task-relevant information. Similarly, in offline RL settings, Mark et al. (2023) experimented using offline data to enable faster exploration in online RL settings, disentangling exploration and exploitation in terms of phases of training. Furthermore, decoupling policies is also considered in multi-arm bandit problems (Avner et al., 2012). Our work not only differ from these works in the aforementioned way of encouraging exploration, but also in that our work is done in pure online RL settings without meta-RL.

**Optimistic state-action value estimates.** One line of work is to integrate ideas from offline-RL algorithms to obtain more optimal Q-value estimates. In particular, (Ji et al., 2024; Luo et al., 2024; Omura et al., 2025) adopted IQL (Kostrikov et al., 2022) to learn the Bellman Optimality Operator, which are known to accelerate training speed albeit being more biased compared to the Bellman SARSA Operator. These works offer different strategies to blend them into policy updates, by either merging the estimates (Ji et al., 2024) or use an annealing schedule (Omura et al., 2025). Our work differs from these line of work, where we trained disentangled policies and use them to obtain mixture of both optimistic and conservative state-action pairs into the replay buffer, whilst they use the conservative replay buffer to estimate optimal values. These works, similar to ours, introduces additional networks for training.

## 7 LIMITATION AND CONCLUSION

This work introduces BOXD, a novel algorithm predicated on the principle of disentangling exploration and exploitation. Our proposed method leverages the established interpretation of dropout as a Bayesian approximation, allowing for the quantification of epistemic uncertainty from the model. Utilising the UCB principle, we propose training a dedicated exploration policy $\pi^{\text{explore}}$ guided by an UCB that may be directly estimated from this uncertainty, thereby enabling a more effective exploration of the state-action space. Furthermore, we propose a strategy to annealing condition which policy to use to interact with the environment. This annealing strategy may improve stability by introducing both exploration and exploitation samples into the replay buffer directly. We have shown that our proposed method significantly outperforms baselines and related works in challenging tasks.

A primary limitation of our proposed BOXD stems from the additional computational costs. This overhead is a direct result of: 1) the $k$ estimates required to compute the maximum due to different dropout masks; 2) maintaining a separate neural network for the optimistic policy $\pi^{\text{explore}}$; and 3) maintaining the corresponding ensemble of $n$ of Q-functions $Q^{\text{explore}}$. Whilst our empirical results demonstrate that effective performance may be achieved with a minimal number of these functions (i.e., k=2, n=2), the introduction of these supplementary networks inevitably increases both memory and computational requirements. One potential future work direction is to incorporate more sophisticated strategies to sample the optimistic and conservative samples from the replay buffer, such as using Prioritised Experience Replay (PER) (Schaul et al., 2015) or Hindsight Experience Replay (HER) (Andrychowicz et al., 2017), in order to further make use of these optimistic and conservative samples obtained by disentangling exploration and exploitation policies. Another potential future work is to blend the optimistic and conservative value estimates directly into updating Q values and build directly upon our work.

## REPRODUCIBILITY STATEMENT

We implement our method in JAX (Bradbury et al., 2018). Details on implementation including the hyperparameters helpful for reproduction of our method are included in Section C. A comprehensive ablation study of design choices may be found in Section D. We have also included the source code used for our experiments in the supplementary material for reference.

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

## A    LARGE LANGUAGE MODELS USAGE DISCLOSURE

We have utilised Large Language Models (LLM) in the writing of this work to help with word polishing and grammar checking.

## B    PRELIMINARY EXPERIMENTS DETAILS

In the preliminary toy experiment in Section 3, we compared SARSA-based updates in actor-critic models using the environment shown in left sub-figure of Figure 2. We compared the optimistic UCB Q-values and the annealing counterpart to the default SARSA-based update. The critic stores the estimated Q-values for each state-action pair in a table and updates them based on either UCB or SARSA. The policy manages logits for each state-action pair in a table, calculates the distribution using the softmax function, and samples actions from this distribution. The policy may be expressed as follows:

$$\pi_\theta(a \mid s) = \frac{\exp(\theta_{s,a})}{\sum_b \exp(\theta_{s,b})} \tag{10}$$

The update of these logits is performed using the policy gradient method, with the update equation given as follows:

$$\begin{aligned} \theta &\leftarrow \theta + \alpha \nabla_\theta \log \pi_\theta(a_t \mid s_t) Q(s_t, a_t), \\ \nabla_{\theta_{s,a'}} \log \pi_\theta(a \mid s) &= \delta_{a,a'} - \pi_\theta(a' \mid s), \end{aligned} \tag{11}$$

where $\delta_{a,a'}$ is the Kronecker delta. The step size $\alpha$ used for updates in both the critic and the policy was set to 5e-4. This step size was chosen because it yields smoother learning curves. Whilst increasing the step size accelerates learning, the observation that UCB-based updates can converge faster than SARSA-based updates remained consistent. The initial state was randomly selected from $s_0$, $s_1$, and $s_2$ with equal probability. Additionally, a probability of 10% of taking random actions, akin to an $\epsilon$-greedy policy, was introduced. We run the toy example for 20 times, and plot the mean and the std of Q-table.

## C    EXPERIMENTS IMPLEMENTATION DETAILS & HYPERPARAMETERS

Our implementation and experiments are done in JAX (Bradbury et al., 2018). Specifically, the versions of important libraries we use in our experiments are: JAX 0.4.30, (Bradbury et al., 2018), MuJoCo 3.3.5 Todorov et al. (2012), Deepmind Control Suite 1.0.31, (Tunyasuvunakool et al., 2020) and gym 0.23.1 (Brockman et al., 2016). Nevertheless, we do not expect a lot of empirical performance even if the library versions do not follow exactly ours.

**Shared across all algorithms**. The replay buffer size is set to $10^6$, and the discount factor $\gamma$ is set to 0.99. The target update rate $\tau$ for target network(s) is 0.005. We have initial random collect steps of 10000. To ensure a fair comparison, all methods employ a batch size of 256, and all neural networks used two hidden layers consisting of 256 units each. All methods use ReLU (Agarap, 2018) as activation function. We use Adam (Kingma & Ba, 2015) as optimiser for all neural networks with the learning rate set to 0.0003.

**TD3, SAC and REDQ**. For baselines, TD3 (Fujimoto et al., 2018), SAC (Haarnoja et al., 2018) and REDQ are both implemented closely following excellent public repositories such as JAXRL https://github.com/ikostrikov/jaxrl, high-replay-ratio (D'Oro et al., 2023) https://github.com/proceduralia/high_replay_ratio_continuous_control and annealing-q-learning (Omura et al., 2025), (https://github.com/motokiomura/annealed-q-learning). We use the default hyperparameters provided in these algorithms. That is, in REDQ, default usage is to randomly select 2 critics from 10 critics to calculated the target $y_i$, whilst updating the critic loss from all 10 critics. For REDQ, all training is using $n = 2$. Additionally, whilst REDQ was originally developed for higher update-to-data (UTD) settings, we set UTD=1 for our experiments.

**BOXD**. Our implementation closely follow DroQ based on SAC and TD3. For each Q-function we add dropout (Srivastava et al., 2014) and layernorm (Ba et al., 2016) consequently after each linear

layer (i.e. linear $\to$ dropout $\to$ layernorm $\to$ activation), except to the last linear layer. For our proposed method, in all experiments and our proposed method, the dropout probability rate set to $p = 0.001$ and is the same across all tasks. we include in Table 2 the tuned number of critics $n$ and number of samples $k$ from the critics is used for each task. However, as shown in Section D, we can achieve good performance by setting to default values $k = 2, n = 2$.

Table 2: numbers of critics $n$ and numbers of samples $k$ used in our experiments.

| Algorithm | TD3+BOXD | | SAC+BOXD | | REDQ+BOXD | |
|---|---|---|---|---|---|---|
| Task | n | k | n | k | n | k |
| cheetah-run | 2 | 2 | 2 | 2 | 2 | 2 |
| finger-turn_hard | 2 | 2 | 2 | 2 | 2 | 2 |
| fish-swim | 2 | 2 | 2 | 2 | 2 | 2 |
| hopper-hop | 10 | 2 | 10 | 2 | 2 | 2 |
| hopper-stand | 2 | 2 | 2 | 2 | 2 | 2 |
| humanoid-run | 2 | 2 | 2 | 3 | 2 | 2 |
| humanoid-stand | 2 | 4 | 2 | 3 | 2 | 2 |
| humanoid-walk | 2 | 2 | 2 | 4 | 2 | 2 |
| quadruped-run | 2 | 10 | 2 | 3 | 2 | 2 |
| quadruped-walk | 2 | 2 | 2 | 3 | 2 | 2 |
| walker-run | 2 | 2 | 2 | 2 | 2 | 2 |

**OAC**. For OAC, which was officially implemented in PyTorch (Paszke et al., 2019), we re-implemented into JAX. For hyperparameters, we followed the publicly released official implementation (https://github.com/microsoft/oac-explore/) and use $\beta_{ub} = 4.66$. Additionally, for OAC (Ciosek et al., 2019), we explored with its UCB-related hyperparameter $\beta_{ub} = \{3, 6\}$, but we did not find significant performance differences. An ablation study on $\beta_{ub} = \{3, 6\}$ is included in Section D.

**DERL framework based algorithm**. For DERL, the original work is experimented in discrete tasks with A2C (Mnih et al., 2016) as base algorithm. We adapt their framework of disentangling exploration and exploitation policies to continuous control, and add intrinsic reward bonuses to the exploration policy. The policies are trained with TD3 and SAC as base algorithm. Specifically, in our experiments we add RND (Burda et al., 2019) as the intrinsic reward, and experimented with various coefficients $= \{0.1, 1.0, 5.0\}$ when adding the bonus to the extrinsic reward (from the environment). Only the best result is presented.

**TOP-like algorithm**. For a TOP-like algorithm (Moskovitz et al., 2021), we use the official implementation of BRO (Nauman et al., 2024), whilst disabling the larger and deeper networks (i.e. not using BroNet but using MLP) to ensure fairer comparison. Furthermore, we employ a batch size of 256 instead of 128 as used by default in BRO. Additionally, we set UTD=1 to be consistent for all algorithms.

**BRO**. We use the official implementation of BRO (Nauman et al., 2024), whilst using all default hyperparameters, and uses UTD=1. Results on some tasks are included in Section E.

## D ABLATION STUDIES

As described in the main manuscript, we introduce two additional hyperparameters in estimating the approximate UCB, namely the number of sampling times $k$ of each Q-function, and the number of Q-functions $n$ to train $\pi^{\text{explore}}$. We first show that generally setting $n = 2, k = 2$ is a good start, and if given more computational costs (e.g. setting $k = 4$) for some tasks we may get further enhanced performance. The number of $n$ also affect optimism, and we show that for some tasks where further optimism is desired, setting $n$ higher may be beneficial. On the other hand, for some tasks, being too optimistic is disadvantageous. We aim to answer the following questions for our ablation studies. Additionally, we performed an ablation on the UCB hyperparameter of OAC (Ciosek et al., 2019).

**Q: Ablation: How is the results for default** $n = 2, k = 2$**?**
**A:** Only a few tasks are significantly affected by even more optimism (i.e. setting higher $k$ or higher $n$), namely hopper-hop. Other tasks are only slightly affected.

We show results using default values of $n = 2$ and $k = 2$. Only for the task hopper-hop we see significant difference. For other tasks, the results are similar with higher $k$ and sometimes decays for higher $n$ (especially for humanoid tasks, where too much optimism will cause degrading performance). In Figure 7 we show the results comparing the averaged result with best-tuned hyperparameters results (left) and averaged result with default hyperparameter ($n = 2$, $k = 2$) (right). In Figure 8 we show the full result on all tasks with default hyperparameter ($n = 2$, $k = 2$).

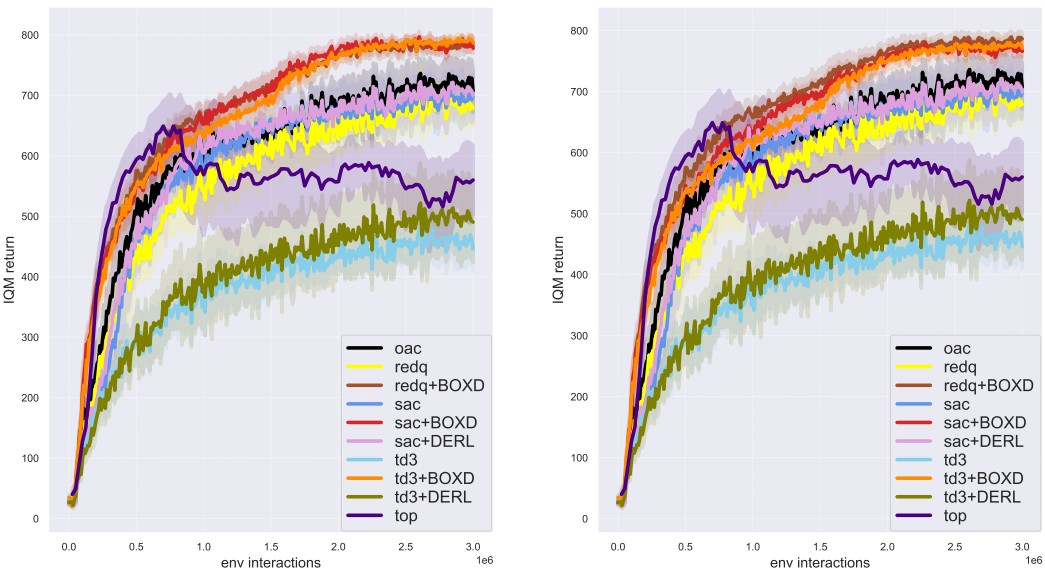

Figure 7: **Left:** Averaged IQM for 11 DM Control tasks of our proposed method with tuned hyperparameters $n$, $k$ versus baselines. **Right:** Averaged IQM of our proposed method using $k = 2$, $n = 2$ versus baselines.

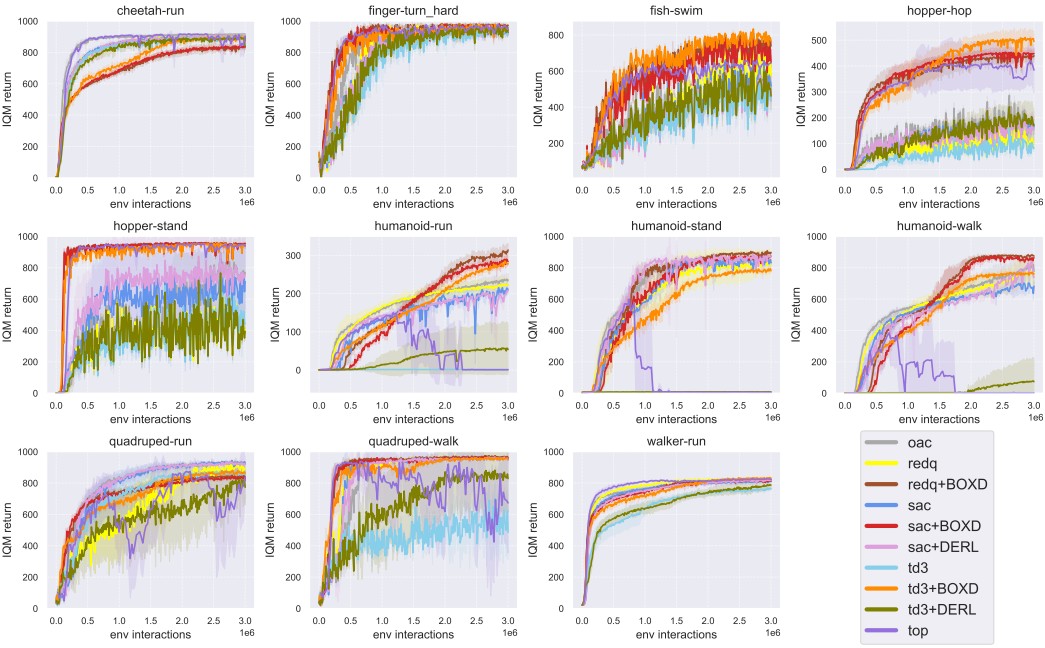

Figure 8: Results for using default hyperparameters $n = 2$, $k = 2$. Generally speaking there is not a significant performance difference compared to tuned hyperparameters version in Figure 4, except for the task hopper-hop.

**Q: Ablation: How does the number of times of Q sampling $k$ affect performance?**
**A:** Depending on the tasks. We recommend to use $k = 2$ as a starting point.

We additionally train $k = \{3, 4, 10\}$ for our algorithm based on TD3 and SAC. Similar trends may be observed in both TD3-based and SAC-based results. Generally speaking, using $k = 2$ is a good start. For tasks that requires more exploration such as hopper-hop, using higher $k$ may result in better performance. Higher $k$ generally does not make the performance decay. The results for TD3-base and the results for SAC-base are shown in Figure 9.

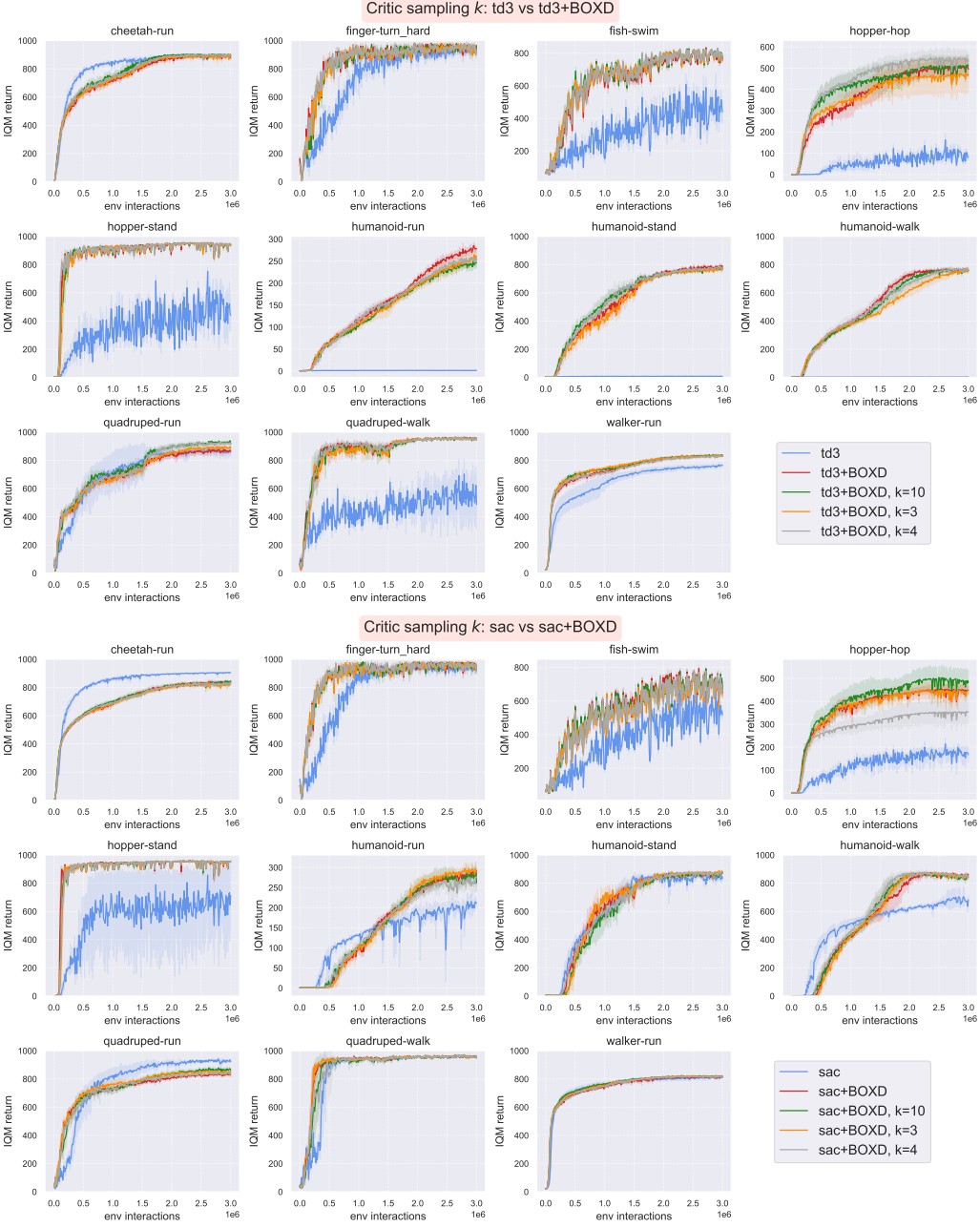

Figure 9: **Top:** Ablation results based on TD3 of the number of critic samples $k$. **Bottom:** Ablation results based on SAC of the number of critic samples $k$. Generally, we do not observe much performance difference when increasing $k$, except hopper-hop where the benefit of higher $k$ is notable.

**Q: Ablation: How does the number of Q-functions $n$ affect performance?**
**A:** Depending on the tasks. Generally speaking, same as contemporary actor-critic algorithms, using $n = 2$ is a good starting point.

The number of critics greatly affects the UCB trade-off parameter $c_n$, as shown in Equation (from (Blom, 1958; Arnold et al., 1992)). We include in Table 3 the approximate values of $c_n$ for $n = \{2, 3, 4, 10\}$. We additionally train $n = \{3, 4, 10\}$ for our algorithm based on TD3 and SAC. For hopper-hop, similarly as using higher $k$, more optimism may be beneficial. However, for humanoid tasks, too much exploration is disadvantageous in both TD3-based SAC-based results. The results for TD3-base and the results for SAC-base are shown in Figure 10.

Table 3: Approximate values for UCB trade-off parameter $c_n$, depending on the number of critics $n$.

| n | 2 | 3 | 4 | 10 |
|---|---|---|---|---|
| $c_n$ | 0.564 | 0.846 | 1.029 | 1.539 |

**Q: How is OAC affected by its UCB hyperparameter $\beta_{ub}$?**
**A**: It does not significantly affect performance.

Generally, the performance is not significantly affected except for the task hopper-hop. Thus we use the official value $\beta_{ub} = 4.66$ for all experiments.

**Q: How does the dropout rate affected performance?**
**A**: In the main text we used a consistent dropout rate of 0.001 for all experiments. This value follows the values investigated by DroQ (Hiraoka et al., 2022), which uses dropout for regularisation. We perform an ablation study with our proposed method BOXD on TD3 on the dropout rate = (0.01, 0.1), using the default optimism related hyperparameters of $n = 2$, $k = 2$. The results are included in Figure 12.

**Q: How effective is the maximum operation?**
**A**: In this work, based on Bayesian principles, we propose to take the maximum of critics coupled with dropout to compute the UCB. One may ask about how optimistic the maximum operation is, and how much performance difference will it make if we only take the mean of the critics, effectively setting the trade-off parameter $c = 0$. We perform ablation study on the comparing the maximum operation that we propose to use versus taking the mean, using the default $n = 2$, $k = 2$. The results are included in Figure 13.

# E    DIRECT COMPARISON WITH BRO

In our results in Section 5 we mainly compared with methods (i.e. OAC, DERL, TOP) that encourages optimism to demonstrate the effectiveness of BOXD. In this section we trained BRO on dm-control tasks, and compared with our proposed method. Furthermore, we also build a version of our proposed BOXD on BRO, named bro+BOXD. Perhaps surprisingly, our proposed BOXD can achieve comparative performance in most tasks whilst having significantly less model capacity. For example, in the hopper-hop task our proposed outperforms BRO. Interestingly, BRO+BOXD does not outperform these built on smaller models, reaching similar performance td3+BOXD, showing that our method does not require deeper and larger networks, thus enable much faster training time. To facilitate easier visualisation, we omit the results of OAC, DERL and TOP in the results are shown in Figure 14.

# F    ANNEALING RESULTS

In the main manuscript, we described our strategy of annealing conditioning to select the action policy $\pi^{\text{act}}$ to interact with the environment. We showed the aggregated average IQM return in the main manuscript, and here we include results on the 11 individual tasks regarding the way annealing policy conditioning is used. Using a 50-50 conditioning strategy is generally good as well. Furthermore, we include another version of pure linear annealing, where instead of Equation (9), we set $s = \frac{t}{T}$ so the threshold $s$ is purely linearly increasing according to current step $t$ versus the total training steps $T$. This ablation is noted as linear annealing in Figure 15.

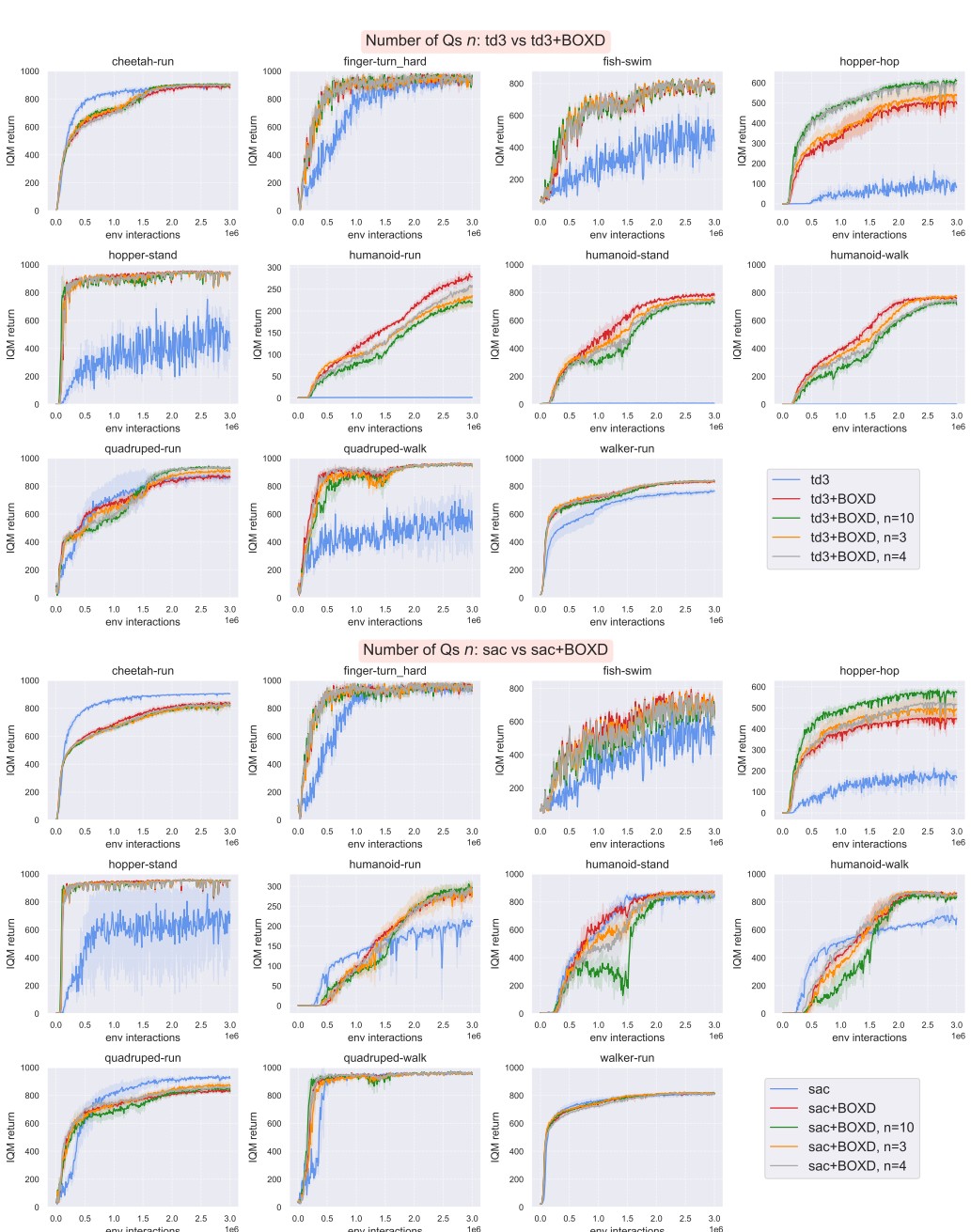

Figure 10: **Top:** Ablation results based on TD3 of the number of Q-functions $n$. **Bottom:** Ablation results based on SAC of the number of Q-functions $n$. Increasing $n$ may induce more optimism, which may be beneficial in some tasks (namely hopper-hop) but disadvantageous for humanoid tasks.

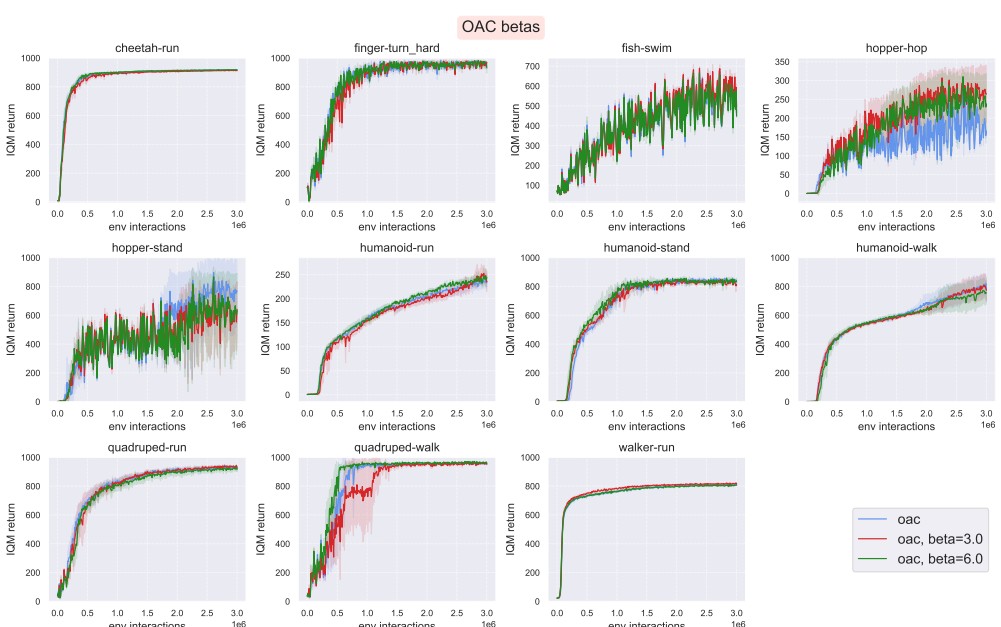

Figure 11: Ablation results based on OAC of its UCB hyperparameter $\beta_{ub}$. Generally, the performances are not really affected.

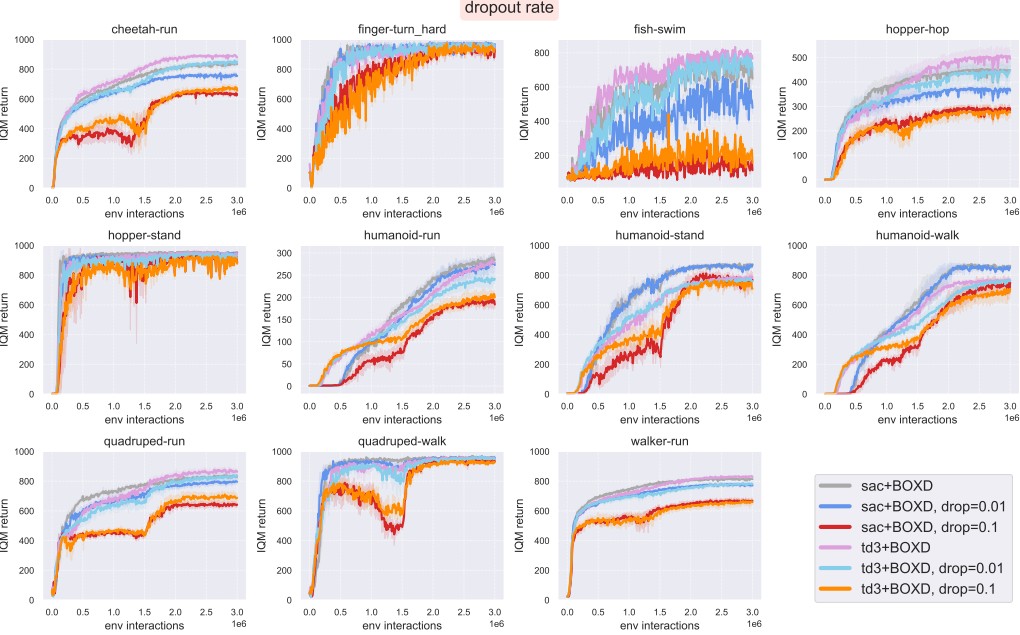

Figure 12: An ablation study on dropout rates in (0.001, 0.01, 0.1). The label without dropout rate is default 0.001. Generally 0.001 dropout rate can achieve stable performance. Increasing too much has some detrimental effect.

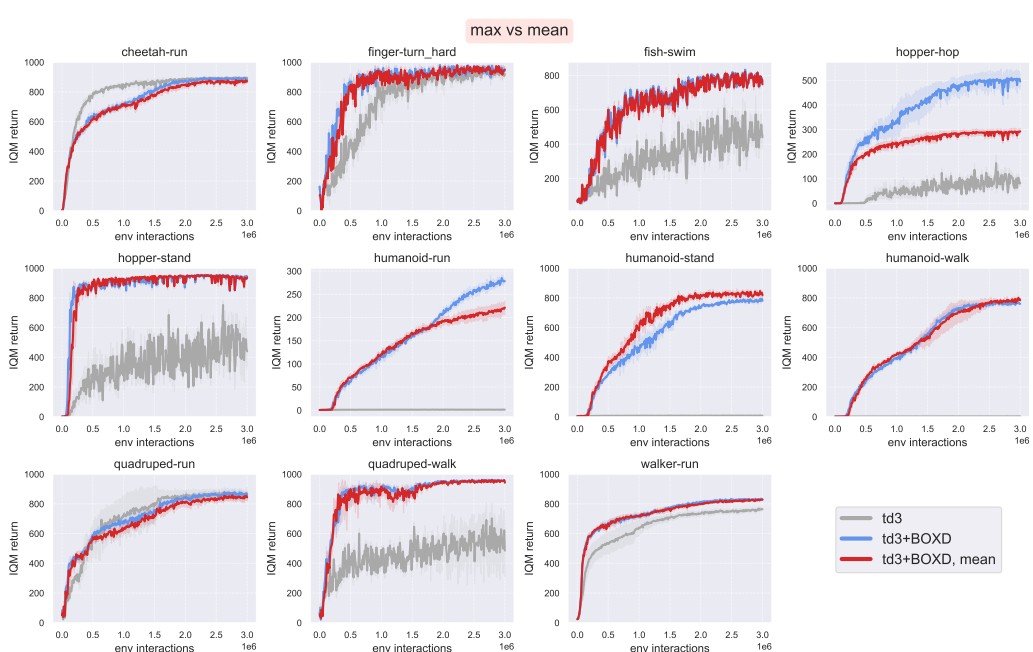

Figure 13: An ablation study on maximum operation. Generally taking the mean is worse then taking the max, which means the optimism is helpful generally for more efficient exploration. Especially in the harder exploration task such as humanoid-run and hopper-hop.

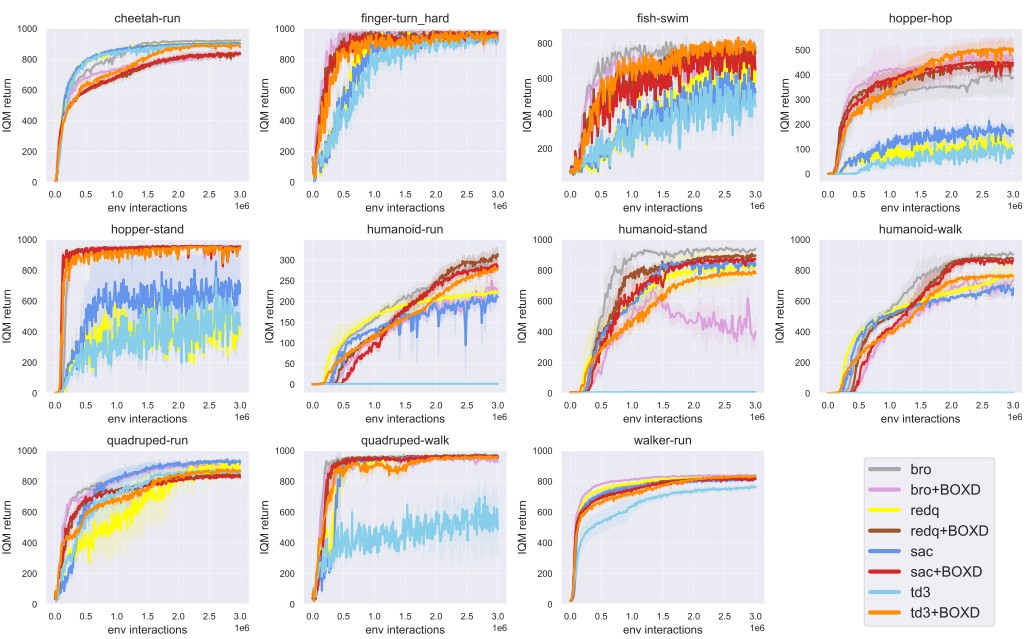

Figure 14: Comparing with BRO, our proposed method can achieve best or near-best performance. Our proposed td3+BOXD, sac+BOXD can achieve similar performance in most tasks, whilst outperforming in some, using approximately half of the computation time compared to BRO Table 1.

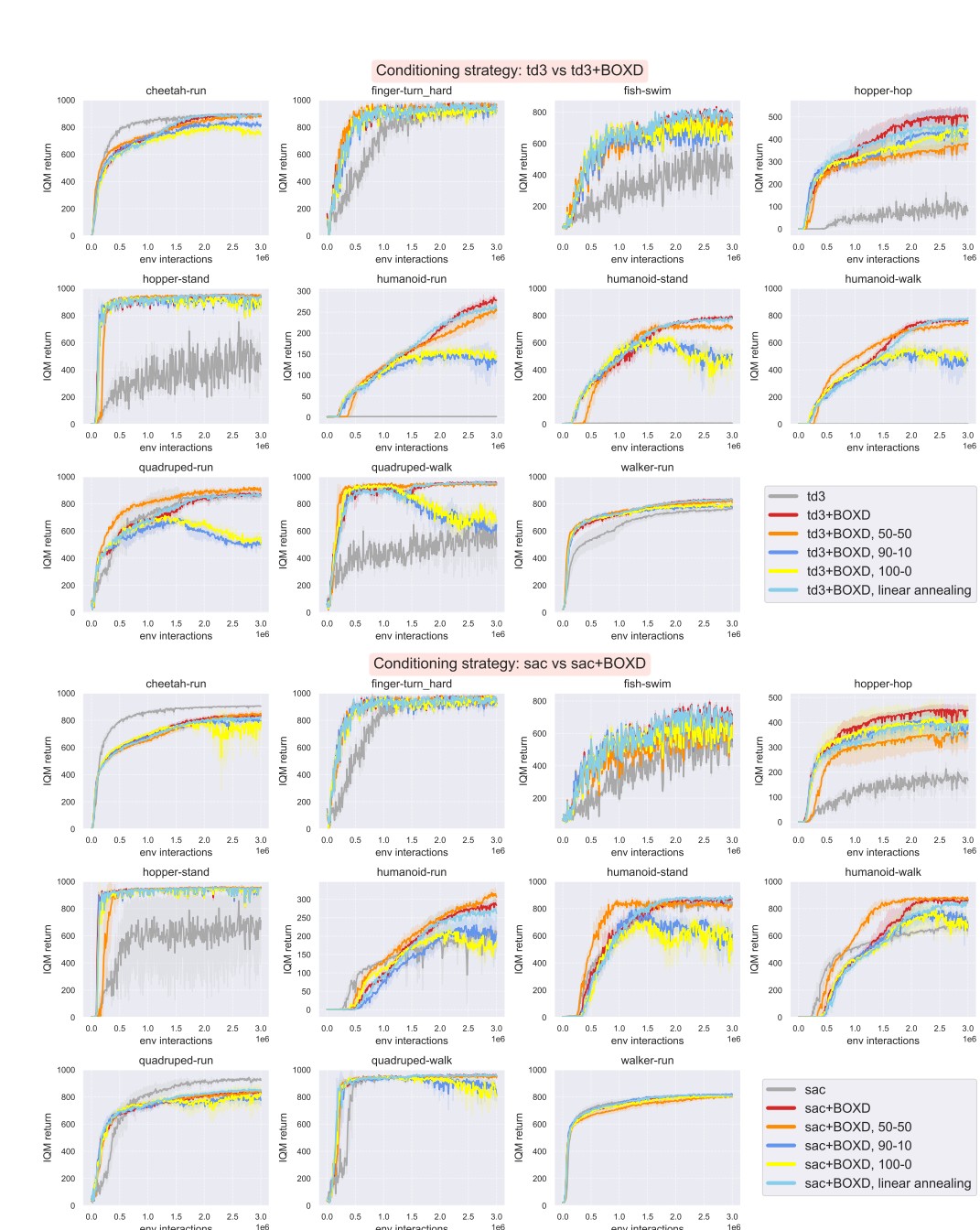

Figure 15: **Top:** Our proposed algorithm with TD3. **Bottom:** Our proposed algorithm with SAC. Our proposed annealing policy conditioning achieves generally the best performance amongst conditioning strategies. Using a fixed conditioning or no conditioning strategy will make the acting policy saturate.

