# OpenReview forum: "Harnessing Bayesian Optimism with Dual Policies in Reinforcement Learning"
_ICLR.cc/2026/Conference — Submitted to ICLR 2026_

### Official Review · Reviewer_CpNo · 2025-10-20

**Soundness:** 2
**Presentation:** 2
**Contribution:** 1
**Rating:** 2
**Confidence:** 4

**Summary:**

The paper proposes BOXD, an online continuous-control RL framework that maintains separate “optimistic” and “task” actor–critic branches. The optimistic branch uses dropout-based critic ensembles whose maximum Q estimates approximate a UCB-style bonus, while the task branch follows standard TD3 or SAC training. Data collection stochastically alternates between the two policies via a linearly annealed schedule so the replay buffer mixes optimistic and conservative samples. On eleven DM Control tasks, BOXD is reported to outperform TD3, SAC, OAC, and a DERL-inspired baseline, with ablations exploring ensemble size, dropout samples, and policy-selection schedules.

**Strengths:**

- Leverages a simple dropout-ensemble max operator to approximate a UCB bonus in continuous control without explicit variance estimation, making the optimism mechanism easy to graft onto SAC/TD3.
- Demonstrates consistent or improved performance over baselines across a subset of DM Control Suite tasks.

**Weaknesses:**

- The proposed “dual policy” story fails to substantiate a better exploration/exploitation balance: both actors write to the same replay buffer and the acting policy is chosen by a fixed linear annealing schedule, so the mechanism collapses to a single mixed policy without adaptivity or evidence of superior exploration; unlike truly dynamic strategies such as Thompson Sampling, the exploration rate never responds to uncertainty or task progress in different regions of the state space.
- Comparisons omit closely related optimistic dual-policy methods (e.g., BRO [1]) and more recent strong model-free baselines (e.g., SimbaV2 [2]), leaving unclear whether BOXD advances state of the art beyond the specific baselines chosen.
- The optimistic claim is not validated on harder or exploration-heavy tasks; the evaluation omits the Dog subset of DM Control and any sparse-reward benchmark (e.g., Maze2D), so the optimistic branch’s benefit remains untested where exploration pressure is high.
- Reporting leaves gaps: the text alternates between IQM and average returns for Figure 4, wall-clock costs versus SAC/TD3 are missing.

[1] Nauman et al., Bigger, Regularized, Optimistic: scaling for compute and sample-efficient continuous control, 2024.

[2] Lee et al., Hyperspherical Normalization for Scalable Deep Reinforcement Learning, 2025.

**Questions:**

- Why are recent state-of-the-art continuous-control baselines—such as BRO [1] and SimbaV2 [2]—absent from the comparison, and how might BOXD perform against them?
- Why were the harder Dog tasks from the DM Control suite omitted, given the claimed exploration improvements?
- Would the authors consider adding experiments on sparse-reward benchmarks (e.g., Maze2D) to showcase the optimistic branch’s exploration benefits?
- In lines 196–200, the authors write that "it may be computationally expensive." What does "it" refer to in this context, and what evidence supports the claimed expense?
- The paper argues that early exploration on humanoid-run ultimately improves asymptotic performance despite an initial lag behind SAC. Could the authors provide analyses or ablations—e.g., varying the proportion of exploration early in training—to demonstrate that the optimistic sampling schedule is responsible for the late-stage gains?
- The annealing rule in Eq. 9 uses a hard-coded factor of 10. Could the authors justify this choice or provide an ablation/sensitivity analysis?
- Do the authors keep all other hyperparameters identical across tasks aside from $(n,k)$? If not, please detail any task-specific tuning.
- Figure 4’s caption and main text alternate between IQM and simple average returns. Could the authors clarify which statistic is reported and update the plots accordingly?
- Please report wall-clock training time relative to SAC/TD3, given the added actor/critic ensembles?

[1] Nauman et al., Bigger, Regularized, Optimistic: scaling for compute and sample-efficient continuous control, 2024.

[2] Lee et al., Hyperspherical Normalization for Scalable Deep Reinforcement Learning, 2025.

---

> ### Author Response · Authors · 2025-11-21
>
> We are grateful for your insightful review and are encouraged that you found our work's strength in easy to append to existing algorithms. We would like to clarify a few issues you pointed out.
>
> ---
>
> **Why are recent state-of-the-art continuous-control baselines—such as BRO [1] and SimbaV2 [2]—absent from the comparison, and how might BOXD perform against them?**
>
> Directly comparing with BRO [1] is not straightforward, since BRO uses (1) much larger and complex NNs (2) higher update-to-data (UTD) ratios which slows down training and (3) single critic for both exploration and exploitation policies. Furthermore, our proposed method is simpler to implement compared to BRO.
> We plan to do some comparison with BRO[1], where we plan to replace the optimistic part of BRO with our algorithm while keeping UTD=1. Specifically, instead of using a trainable $\beta$ in BRO (lines 11, 12, 14, 15 of Algorithm 2, in the page 17 of arxiv version of their paper), we replaced it with our proposed method. We will update the manuscript to include the training step v.s. IQM plot together at an appropriate time.
>
> ---
>
> **Dog tasks**
>
> Thank you for the suggestion. We have included the 4 dog tasks compared to baselines SAC and TD3, and show the final performance below. We will update the manuscript to include the training step v.s. IQM plot together at an appropriate time.
>
> | algorithms | tasks     | IQM   | IQM-std |
> |------------|-----------|-------|---------|
> | td3        | dog-run   | 180.1 | 12.9    |
> | td3_boxd2  | dog-run   | 289.6 | 15.6    |
> | sac        | dog-run   | 6.0   | 1.1     |
> | sac_boxd2  | dog-run   | 337.4 | 35.6    |
> | td3        | dog-stand | 669.4 | 57.7    |
> | td3_boxd2  | dog-stand | 911.1 | 7.5     |
> | sac        | dog-stand | 308.9 | 326.1   |
> | sac_boxd2  | dog-stand | 944.3 | 10.6    |
> | td3        | dog-trot  | 409.3 | 31.8    |
> | td3_boxd2  | dog-trot  | 559.6 | 24.4    |
> | sac        | dog-trot  | 8.4   | 2.0     |
> | sac_boxd2  | dog-trot  | 778.1 | 11.0    |
> | td3        | dog-walk  | 743.0 | 62.3    |
> | td3_boxd2  | dog-walk  | 859.7 | 7.4     |
> | sac        | dog-walk  | 257.1 | 239.6   |
> | sac_boxd2  | dog-walk  | 888.5 | 11.5    |
>
> ---
>
> **sparse-reward benchmarks (e.g., Maze2D) **
>
> Thank you for this suggestion. We have trained on two point-maze tasks in gymnasium-robotics [3], using our proposed method versus baselines while additionally including BRO. As shown, for harder sparse-reward tasks (large maze) our proposed algorithm can outperform baselines significantly and for easier sparse-reward tasks (medium maze) the performance is similar to baseline. If you have some other specific task in mind, we would be happy to run on.
>
>
> | algorithms | tasks               | IQM   | IQM-std |
> |------------|---------------------|-------|---------|
> | td3        | PointMaze_Medium-v3 | 486.4 | 27.8    |
> | td3_boxd2  | PointMaze_Medium-v3 | 496.2 | 18.9    |
> | sac        | PointMaze_Medium-v3 | 511.8 | 8.7     |
> | sac_boxd2  | PointMaze_Medium-v3 | 486   | 22.1    |
> | td3        | PointMaze_Large-v3  | 0.0   | 0.0     |
> | td3_boxd2  | PointMaze_Large-v3  | 494.8 | 250.1   |
> | sac        | PointMaze_Large-v3  | 55.4  | 110.8   |
> | sac_boxd2  | PointMaze_Large-v3  | 362.2 | 297.4   |
>
> ---
>
> **In lines 196–200, the authors write that "it may be computationally expensive." What does "it" refer to in this context, and what evidence supports the claimed expense?**
>
> For example in BRO and OAC, the optimistic operation is done by training/tuning the $\beta$ hyperparameter in UCB. BRO, inspired by TOP[4], introduced two more additional networks to learn this optimistic hyperpameter. We show that we can skip this optimisation step, which can be expensive, by taking the maximum of Q functions.  We will reflect this clearly in the manuscript.
>
> ---
>
> **provide analyses or ablations—e.g., varying the proportion of exploration early in training—to demonstrate that the optimistic sampling schedule is responsible for the late-stage gains?**
>
> We apologise that we did not fully understand the suggestion. We are not really sure how to follow the suggested scheme. Our proportion of exploration stems from the maximum operation and a weighted-random choosing the exploration policy, rather than the tunable/trainable UCB parameter as has been done in related works (e.g. OAC, BRO). We have done ablation on the policy conditioning scheme (i.e. different ways to determine which policy to choose) in Appendix E.

---

> > ### Author Response · Authors · 2025-11-21
> >
> > ---
> >
> > **The annealing rule in Eq. 9 uses a hard-coded factor of 10. Some ablation?**
> >
> > We apologise for the confusion. This factor is simply used to ease calculating the threshold (for humans, i.e. threshold=0.1,0.2,0.3, …1) of sampling between exploration and task policies. We train a purely linear schedule (i.e. the threshold increases from 0 to 1 purely linearly by setting the threshold=t/T) which completely removes the need for Equation 9. The results alongside the proposed conditioning scheme in Equation 9 are shown below. As shown, there is no significant difference between the proposed scheme and a pure linear schedule. We will reflect this on the annealing strategy ablation in the Appendix E when we update the manuscript.
> >
> > | algorithms                | tasks            | IQM   | IQM-std |
> > |---------------------------|------------------|-------|---------|
> > | td3_boxd2                 | cheetah-run      | 879.2 | 13.5    |
> > | td3_boxd2_linearannealing | cheetah-run      | 890.8 | 4.4     |
> > | td3_boxd2                 | finger-turn_hard | 942.4 | 24.7    |
> > | td3_boxd2_linearannealing | finger-turn_hard | 955.1 | 15.5    |
> > | td3_boxd2                 | fish-swim        | 765.8 | 7.7     |
> > | td3_boxd2_linearannealing | fish-swim        | 770.6 | 8.9     |
> > | td3_boxd2                 | hopper-hop       | 495.0 | 30.7    |
> > | td3_boxd2_linearannealing | hopper-hop       | 464.9 | 82.6    |
> > | td3_boxd2                 | hopper-stand     | 937.5 | 13.6    |
> > | td3_boxd2_linearannealing | hopper-stand     | 947.1 | 2.4     |
> > | td3_boxd2                 | humanoid-run     | 278.3 | 4.4     |
> > | td3_boxd2_linearannealing | humanoid-run     | 259.3 | 14.4    |
> > | td3_boxd2                 | humanoid-stand   | 780.6 | 10.1    |
> > | td3_boxd2_linearannealing | humanoid-stand   | 777.5 | 16.9    |
> > | td3_boxd2                 | humanoid-walk    | 760.3 | 4.5     |
> > | td3_boxd2_linearannealing | humanoid-walk    | 782.8 | 6.6     |
> > | td3_boxd2                 | quadruped-run    | 862.9 | 18.6    |
> > | td3_boxd2_linearannealing | quadruped-run    | 864.0 | 23.1    |
> > | td3_boxd2                 | quadruped-walk   | 951.8 | 6.0     |
> > | td3_boxd2_linearannealing | quadruped-walk   | 946.6 | 5.1     |
> > | td3_boxd2                 | walker-run       | 829.3 | 5.6     |
> > | td3_boxd2_linearannealing | walker-run       | 828.8 | 3.0     |
> > | sac_boxd2                 | cheetah-run      | 834.0 | 13.8    |
> > | sac_boxd2_linearannealing | cheetah-run      | 819.6 | 18.9    |
> > | sac_boxd2                 | finger-turn_hard | 924.6 | 0.5     |
> > | sac_boxd2_linearannealing | finger-turn_hard | 950.0 | 21.3    |
> > | sac_boxd2                 | fish-swim        | 648.5 | 30.9    |
> > | sac_boxd2_linearannealing | fish-swim        | 671.7 | 14.5    |
> > | sac_boxd2                 | hopper-hop       | 448.4 | 18.8    |
> > | sac_boxd2_linearannealing | hopper-hop       | 392.2 | 67.7    |
> > | sac_boxd2                 | hopper-stand     | 954.6 | 2.0     |
> > | sac_boxd2_linearannealing | hopper-stand     | 950.7 | 0.9     |
> > | sac_boxd2                 | humanoid-run     | 284.0 | 8.0     |
> > | sac_boxd2_linearannealing | humanoid-run     | 261.4 | 22.2    |
> > | sac_boxd2                 | humanoid-stand   | 873.2 | 5.7     |
> > | sac_boxd2_linearannealing | humanoid-stand   | 879.3 | 5.6     |
> > | sac_boxd2                 | humanoid-walk    | 863.5 | 4.7     |
> > | sac_boxd2_linearannealing | humanoid-walk    | 848.9 | 8.8     |
> > | sac_boxd2                 | quadruped-run    | 828.1 | 14.8    |
> > | sac_boxd2_linearannealing | quadruped-run    | 842.9 | 12.0    |
> > | sac_boxd2                 | quadruped-walk   | 951.5 | 4.1     |
> > | sac_boxd2_linearannealing | quadruped-walk   | 952.9 | 2.9     |
> > | sac_boxd2                 | walker-run       | 813.2 | 2.0     |
> > | sac_boxd2_linearannealing | walker-run       | 809.7 | 5.1     |
> >
> > ---
> >
> > **Do the authors keep all other hyperparameters identical across tasks aside from (n,k) ? If not, please detail any task-specific tuning.**
> >
> > Thank you for this important question regarding reproductivity. We would like to confirm that all hyperparameters, except those introduced additionally by our work (i.e. n and k) are identical across all tasks for all algorithms, which are also the same as their respective (baseline) work.
> >
> > ---
> >
> > **main text alternate between IQM and simple average returns. Could the authors clarify which statistic is reported and update the plots accordingly?**
> >
> > We would like to confirm that all results throughout the manuscript are IQM. We apologise for the confusion and will update the legends accordingly when we update the manuscript.
> >
> > ---
> >
> > **wall-clock training time**
> >
> > We will perform wallclock comparison with other algorithms raised by other reviewers, including BRO, and report back in a later date.
> >
> > ---
> >
> > [3] Rodrigo de Lazcano et al., Gymnasium Robotics, http://github.com/Farama-Foundation/Gymnasium-Robotics
> > [4] Moskovitz et al., Tactical Optimism and Pessimism for Deep Reinforcement Learning, NeurIPS, 2021

---

### Official Review · Reviewer_RzfA · 2025-10-23

**Soundness:** 2
**Presentation:** 3
**Contribution:** 1
**Rating:** 2
**Confidence:** 5

**Summary:**

The paper addresses the exploration-exploitation dilemma of reinforcement learning in a continuous control context. The suggested approach trains separate policies for each of these conflicting purposes by treating the maximum critic response in Bellman target estimation for exploration and minimum as in the standard practise for exploitation. The suggested recipe has been appended to two mainstream base models, SAC and TD3, and has been tested on a number of continuous control tasks taken from the DeepMind Control Suite.

**Strengths:**

The main results reported in Figure 4 as well as the more detailed results provided in the appendix indicate a trend in favor of the suggested approach. Particularly, the approach consistently improves performance when dropped into a base learner such as SAC or TD3.

**Weaknesses:**

None of the suggested contributions listed in Lines 85-98 are novel:
 * Epistemic uncertainty quantification with Monte Carlo dropout is a rather addendum as also suggested in DroQ.
 * Calculating UCB/LCB over multiple Q-functions has been extensively studied in prior work such as OAC cited by the authors and EDAC [1] in the offline RL context, the online application of which is straightforward. Neutralizing the over-pessimism of critic ensembles also has a thick literature even with algorithms that can tune its degree dynamically in the course of training. For example see [2,3] and their references. For example, the TOP algorithm [3] performs Bayesian optimization to dynamically tune the degree of optimism and pessimism in actors and critics, respectively.
 * I also expect the shown performance improvements to shrink when the method is appended to REDQ [4], which is the established state of the art in performing SAC with more than two critics
 *  To my take BOXD is only a special case of many of the more advanced algorithms suggested above and its demonstrated improvement stems from a model capacity extension, e.g. introducing more free parameters, which can also be done in arbitrary other ways.

It is a big weakness that the paper does not recognize these contributions, does not differentiate its solution from them, and does not provide quantitative comparison against them.

Lines 468-471 conclude the paper by presenting the novelty as *"we propose training a dedicated exploration policy $\pi^{\text{explore}}$ guided by UCB that may be directed estimated from this uncertainty ..."*. I am not able point out the novelty here what is already suggested in OAC and [1,2,3], at least conceptually. This may either be due to the lack of novelty or a too high-level description of it in the paper. In both cases, a major revision is essential before the publication of the work.

[1] An et al., Uncertainty-Based Offline Reinforcement Learning
with Diversified Q-Ensemble, NeurIPS, 2021

[2] Cetin and Celiktutan, Learning Pessimism for Robust and Efficient Off-Policy Reinforcement Learning, AAAI, 2023

[3] Moskovitz et al., Tactical Optimism and Pessimism for Deep Reinforcement Learning, NeurIPS, 2021

[4] Chen et al., Randomized Ensembled Double Q-Learning: Learning Fast Without a Model, ICLR, 2021

**Questions:**

* Can the authors confirm that in all plots where the models are compared, the suggested BOXD extension does not use more model capacity than the baselines, be it additional critic copies, a new network trained for exploration purposes and so on? If this is indeed the case, then we need to compare it with the plain baselines whose model capacities are also extended proportionally. Only in this way we can assess the value added of the contribution over sheer capacity enhancement. I wonder the answer to this question also because of Line 475 that puts forward computational cost as a limitation. It is fine if this overhead stems from extra computations made on a fixed hypothesis space. But if it stems from a capacity extension, I have to be sure that the baselines are fairly treated.

 * From Lines 320-322, it reads like DERL has many conceptual similarities to the suggested BOXD. Can the authors detail in which ways BOXD and DERL are similar and what is BOXD doing differently from it? I can see in Figure 4 that DERL tends to match better with TD3 than SAC, which is expected from a reward shaping approach.

---

> ### Author Response · Authors · 2025-11-21
>
> We would like to thank you for your valuable comments. Below, we provide further clarification on the issues mentioned in your review.
>
> ---
>
> **Calculating UCB/LCB over multiple Q-functions has been extensively studied in prior work**
>
> We would like to clarify that one of our main differences with previous related algorithms is that we show that by taking the $\textit{maximum}$ of ensembles, combined with dropout, we can implicitly tune the degree of optimism rather than some sort of dynamic training/tuning as have done in EDAC[1], GPL[2], TOP[3] and BRO[5] which another reviewer pointed out. BRO[5] specifically uses TOP[3] as their optimism structure, and we plan to compare with BRO directly, and we will report back in a later date.
>
> ---
>
> **REDQ**
>
> We would like to clarify that, to our understanding, REDQ[1] is mostly designed to address issues with high update-to-data (UTD) ratio. There are two main differences between BOXD and REDQ. Firstly, our proposed method does not address high UTD ratio settings (i.e. we use UTD=1). Secondly, for the task branch, as mentioned in lines 178-183 of the manuscript, we have only 2 Q functions (i.e. for REDQ’s hyperparameter, n=m=2). That said, we plan to do some preliminary experiments on REDQ and we will report back at a later date.
>
> ---
>
> **BOXD's improvement stems from a model capacity extension**
>
> While we do think that increasing model capacity is an important direction for scaling up RL algorithms, we respectfully disagree with the assessment that naively introducing more free parameters can result in better performance. Simply using larger NNs (e.g. setting hidden dims to for example 32768 or having more layers) would not outperform typically those set to smaller ones [6][7] (e.g. 256 as has been done in this work and most related works).
>
> Instead, following the suggestion of other reviewers, we plan to add an additional baseline as in BRO [8], where various techniques were added, where clever use of larger NNs are employed regarding your comments concerning model capacity.
>
> ---
>
> **novelty compared to OAC**
>
> We acknowledge the similarities between OAC and our proposed method. However, our approach to approximating UCB is distinctly different. Our main contribution is showing that taking the maximum of Q functions, coupled with dropout, can turn into a Bayesian-principle NNs and thus can be used to approximate UCB, rather than using a hand-picked hyperparameter for UCB as done in OAC or some trainable optimistic parameter, as done in BRO. We will revise the manuscript to clearly explain both the similarities and the differences between OAC and our method.
>
> ---
>
> **BOXD uses more model capacity**
>
> As mentioned around lines 178-183 in our manuscript, we have indeed added more models over the baselines. We introduce additional copies of critics and an additional exploration policy to disentangle the exploration and exploitation processes. Therefore, we acknowledge that from the model capacity (i.e. number of trainable parameters) point-of-view there are more parameters to be trained. Suggested by another reviewer, we plan to compare our proposed algorithm to BRO, which uses much larger NNs concerning your concerns about fair model capacity.
>
> ---
>
> **difference to DERL**
>
> Thank you for this important question. DERL is a previous work that proposes to use two distinct sets of policies, an exploration policy and an exploitation policy, in addition to two distinct sets of critics. In this sense, it is conceptually similar to our proposed method.
> However, the main distinct difference is the way exploration is encouraged (i.e. how the exploration policy is trained). DERL uses exploration bonuses such as RND, as we have explained towards the end of the appendix C of our manuscript. Our proposed BOXD uses UCB principles by taking the maximum of Q functions coupled with dropout, derived from Bayesian principles. This is distinctly different from previous works related to optimistic UCB. We will reflect in the relevant section for clarity when we update the manuscript.
>
> ---
>
> [5] M. Nauman et al., “Bigger, Regularized, Optimistic: scaling for compute and sample-efficient continuous control”, NeurIPS 2024
> [6] K. Ota et al., “A framework for training larger networks for deep Reinforcement learning”, Machine Learning 2024
> [7] G. Sokar et al., “Mind the GAP! The Challenges of Scale in Pixel-based Deep Reinforcement Learning”,  NeurIPS 2025

---

> > ### Comment · Reviewer_RzfA · 2025-11-26
> >
> > Thanks for the answer. Then I conclude that all my main concerns leading to a grade below the acceptance threshold are valid and they do not stem from a misunderstanding.
> >
> > I do not think REDQ's being tested in the original paper in the UTD>1 setting makes a conceptual difference to our discussion here. The experiment setup should have been chosen in the setup where the existing alternatives work best. In a hypothetical case where REDQ doesn't suffer from any problems that are posed as a problem in this paper, then the usefulness of the proposed methodology becomes questionable.
> >
> > Now that BOXD is confirmed to use more model capacity, then it is clear where the performance improvement is coming from. My related original comment is still valid.
> >
> > We are on the same page about the comments related to OAC and DERL. My related concerns should be addressed by an update in the manuscript.
> >
> > To wrap up, the author response confirms the gravity of my initial concerns. I keep my score and will be following the updates regarding new experiments.

---

### Official Review · Reviewer_n68U · 2025-10-29

**Soundness:** 2
**Presentation:** 2
**Contribution:** 1
**Rating:** 2
**Confidence:** 4

**Summary:**

The authors proposed BOXD, a method for mediating exploration-exploitation tradeoff, by instantiating separate "conservative" policies that optimize the return given task-dependent rewards, and "exploration" policy that maximize the upper confidence bound (UCB) given the set of value estimates from an ensemble of value networks. An annealing scheme is proposed from exploration to greedy over time and experience with the task. The proposed method is evaluated on continuous control tasks from the dm-control suite.

**Strengths:**

- The paper is clearly written and easy to follow.
- Empirical evaluation show promising results.

**Weaknesses:**

- The idea of independent explorative and exploitative agents for exploration-exploitation tradeoff has been studied in Yu et al., 2022, and the presented study bears strong resemblance to this previous study. The missing citation and relevant discussion undermines the validity and novelty of the current paper.
- The paper's claim to "harnessing Bayesian optimism" is a significant overstatement. The link between the algorithm and a principled UCB and computational principles behind Bayesian optimization is weak, and lack of theoretical justification.
- The proposed method is expected to introduce significant computational overhead. The absent complexity analysis is concerning.

References.

[1] Yu, C., Mguni, D., Li, D., Sootla, A., Wang, J. and Burgess, N., 2022. SEREN: Knowing When to Explore and When to Exploit. arXiv preprint arXiv:2205.15064.

**Questions:**

See above.

---

> ### Author Response · Authors · 2025-11-21
>
> Thank you for the constructive and insightful feedback and find our work easy to follow. We would like to clarify a few points pointed out by your review below.
>
> ---
>
> **The idea of independent explorative and exploitative agents for exploration-exploitation tradeoff has been studied in Yu et al., 2022, and the presented study bears strong resemblance to this previous study. The missing citation and relevant discussion undermines the validity and novelty of the current paper.**
>
> Thank you for raising this point. We would like to clarify that, in the first part of the related work section (Section 6, lines 439-451), we have discussed extensively about relevant works that use independent explorative and exploitative policies. While we have not referred to the particular work [1] as mentioned in your review, we have discussed the differences between our works and related ones in the same literature. We will add this reference to the related works section when we update the manuscript.
>
> ---
>
> **The paper's claim to "harnessing Bayesian optimism" is a significant overstatement. The link between the algorithm and a principled UCB and computational principles behind Bayesian optimization is weak, and lack of theoretical justification.**
>
> We apologise if we have not conveyed the relationship between our proposed method and UCB more clearly. We would like to clarify that we are indeed using the Bayesian framework, using the theoretical framework established by [2], which we also cite in our current manuscript. This is done by enabling dropout during inference, which turns the NNs into an approximate Bayesian model [2]. We then utilise this theoretical framework to derive the approximate optimistic UCB by taking the maximum of Q functions.
>
> ---
>
> **The proposed method is expected to introduce significant computational overhead.**
>
> We acknowledge that we have introduced additional computational overhead compared to baselines. We will perform wallclock comparison with other algorithms raised by other reviewers and report back in a later date.
>
> ---
>
> [2] Yarin Gal and Zoubin Ghahramani, “Dropout as a Bayesian Approximation: Representing Model Uncertainty in Deep Learning”, ICML 2016

---

### Official Review · Reviewer_jL2p · 2025-10-30

**Soundness:** 2
**Presentation:** 2
**Contribution:** 2
**Rating:** 2
**Confidence:** 4

**Summary:**

This work addresses a challenge in deep-learning exploration methods, that a focus on exploration using optimistic value functions can distract the agent from solving the task, while aiming for accurate, conservative value functions may reduce exploratory behavior and lead to inefficient data gathering. The authors propose training two separate Q estimates, and two separate policies, for exploration and for exploitation. The exploratory policy maintains optimism through bootstrapping with UCB value estimates computed through an ensemble of dropout Q networks. Using this method improves the performance of both SAC and TD3 on a suite of continuous control tasks.

**Strengths:**

The derivation of a UCB result from EVT was interesting, and using dropout and ensembles to generate the required standard deviation is clever. The results do significantly improve the baselines. I have not seen using the max over an ensemble to produce optimistic targets.

**Weaknesses:**

The authors do acknowledge this, but the novelty of separating exploration and exploitation is limited, as it has been done many times before. Besides the examples listed in the paper, [1] does this explicitly, and [2] learns a family of bonus-based exploration algorithms with different bonus scales (including b=0).

The EVT math is interesting, however as far as I can tell the variance from dropout is not explicitly used. So in practice, this reduces to “max over ensembles.” In addition, I doubt that dropout_rate=0.001 actually leads to significant variance. Strong performance with 2 networks and 2 samples similarly suggests that the optimistic UCB is not necessarily the important improvement.

Though I have not seen this specific use of optimistic targets using ensembles, a large variety of work has covered using ensembles for bayesian upper bounds on value, such as Bootstrap DQN [3] and SUNRISE [4].

Though the math presented in this paper is interesting, it does not feel strongly connected to the empirical results, and I believe the separate ingredients of ensembles, dropout, and disentangled exploration/exploitation policies have all been thoroughly explored before, if not their combinations.

One presentation issue I wanted to bring up: I believe Figure 6 is the same plot side-by-side?


[1] “Decoupled Exploration and Exploitation Policies for Sample-Efficient Reinforcement Learning”, Whitney et al, https://arxiv.org/pdf/2101.09458
[2] “Never Give Up: Learning directed exploration strategies” Badia et al, https://arxiv.org/pdf/2002.06038
[3] “Deep Exploration via Bootstrapped DQN” Osband et al, https://arxiv.org/abs/1602.04621
[4] “SUNRISE: a simple unified framework for ensemble learning in deep reinforcement learning”, Lee et al, https://arxiv.org/abs/2007.04938

**Questions:**

Do the authors have results clarifying the role of dropout in this method? Does it operate as a training regularizer or something more directly connected to bayesian optimism? Is this method sensitive to dropout coefficient?

Similarly for ensembles, can the authors separate the stability improvements of ensembles from the bayesian optimism? For example, would we expect a “mean over ensemble” to do significantly worse than “max over ensemble”?

---

> ### Author Response · Authors · 2025-11-21
>
> Thank you for your thoughtful and constructive review. We’d like to respond a few points brought up in your review.
>
> ---
>
> **One presentation issue I wanted to bring up: I believe Figure 6 is the same plot side-by-side?**
>
> We apologise for the confusion. We would like to clarify that they are NOT the same plot. The left plot is the tuned performance from tuning parameters n and k for each task, and the right plot is the performance where a minimal version of n=2,k=2 for all tasks is used. We will update the captions of Figure 6 for further clarity. We are quite happy these two subplots are very much similar, so the minimal version is performant enough for most tasks.
>
> ---
>
> **Role of dropout and BOXD's sensitivity to dropout coefficient?**
>
> We thank the reviewer for this critical question. We would like to clarify that, from a theoretical standpoint, the usage of dropout is necessary. Enabling dropout during inference turns the NNs into an approximate Bayesian model (which stems from the work in [5]), not just as a regularisation technique as done in previous works in RL (i.e. DroQ [6]). This addition of dropout enables us to take the maximum directly to approximate UCB without using the variance between ensembles. We would like to emphasise that this is one of the important differences between our work and previous works. We would like to further clarify that the reason we used 0.001 as default is because we mainly followed the values from DroQ [6].
>
> We have done additional ablation on the dropout rate of {0.1, 0.01} and we include the final performance at 3M steps below. As shown, taking a higher dropout rate results in diminished return in most tasks. The first column of each task is using default 0.001 dropout rate. (Note below that for humanoid-walk dropout=0.1 for td3_boxd2 is missing. We will add them shortly.) We will update the manuscript to include the training step v.s. IQM plot together at an appropriate time.
>
> | algorithms         | tasks            | IQM   | IQM-std |
> |--------------------|------------------|-------|---------|
> | td3_boxd2          | cheetah-run      | 879.2 | 13.5    |
> | td3_boxd2_drop0.01 | cheetah-run      | 851.7 | 15.6    |
> | td3_boxd2_drop0.1  | cheetah-run      | 656.4 | 12.6    |
> | td3_boxd2          | finger-turn_hard | 942.4 | 24.7    |
> | td3_boxd2_drop0.01 | finger-turn_hard | 959.8 | 18.1    |
> | td3_boxd2_drop0.1  | finger-turn_hard | 935.6 | 35.1    |
> | td3_boxd2          | fish-swim        | 765.8 | 7.7     |
> | td3_boxd2_drop0.01 | fish-swim        | 700.8 | 6.7     |
> | td3_boxd2_drop0.1  | fish-swim        | 155.8 | 23.7    |
> | td3_boxd2          | hopper-hop       | 495.0 | 30.7    |
> | td3_boxd2_drop0.01 | hopper-hop       | 448.9 | 42.9    |
> | td3_boxd2_drop0.1  | hopper-hop       | 275.9 | 23.1    |
> | td3_boxd2          | hopper-stand     | 937.5 | 13.6    |
> | td3_boxd2_drop0.01 | hopper-stand     | 918.1 | 17.0    |
> | td3_boxd2_drop0.1  | hopper-stand     | 877.9 | 8.5     |
> | td3_boxd2          | humanoid-run     | 278.3 | 4.4     |
> | td3_boxd2_drop0.01 | humanoid-run     | 241.9 | 10.8    |
> | td3_boxd2_drop0.1  | humanoid-run     | 200.1 | 6.7     |
> | td3_boxd2          | humanoid-stand   | 780.6 | 10.1    |
> | td3_boxd2_drop0.01 | humanoid-stand   | 759.7 | 23.8    |
> | td3_boxd2_drop0.1  | humanoid-stand   | 709.4 | 34.7    |
> | td3_boxd2          | humanoid-walk    | 760.3 | 4.5     |
> | td3_boxd2_drop0.01 | humanoid-walk    | 763.1 | 6.3     |
> | td3_boxd2          | quadruped-run    | 862.9 | 18.6    |
> | td3_boxd2_drop0.01 | quadruped-run    | 835.5 | 28.9    |
> | td3_boxd2_drop0.1  | quadruped-run    | 688.1 | 22.0    |
> | td3_boxd2          | quadruped-walk   | 951.8 | 6.0     |
> | td3_boxd2_drop0.01 | quadruped-walk   | 947.9 | 3.1     |
> | td3_boxd2_drop0.1  | quadruped-walk   | 928.0 | 5.7     |
> | td3_boxd2          | walker-run       | 829.3 | 5.6     |
> | td3_boxd2_drop0.01 | walker-run       | 779.6 | 15.5    |
> | td3_boxd2_drop0.1  | walker-run       | 659.0 | 17.9    |

---

> ### Author Response · Authors · 2025-11-21
>
> *Continued for dropout ablation*
>
> | algorithms         | tasks            | IQM   | IQM-std |
> |--------------------|------------------|-------|---------|
> | sac_boxd2          | cheetah-run      | 834.0 | 13.7    |
> | sac_boxd2_drop0.1  | cheetah-run      | 632.8 | 4.0     |
> | sac_boxd2_drop0.01 | cheetah-run      | 754.6 | 5.0     |
> | sac_boxd2          | finger-turn_hard | 924.6 | 0.5     |
> | sac_boxd2_drop0.01 | finger-turn_hard | 969.2 | 2.2     |
> | sac_boxd2_drop0.1  | finger-turn_hard | 934.8 | 14.3    |
> | sac_boxd2          | fish-swim        | 648.5 | 30.9    |
> | sac_boxd2_drop0.01 | fish-swim        | 477.9 | 34.1    |
> | sac_boxd2_drop0.1  | fish-swim        | 112.7 | 1.3     |
> | sac_boxd2          | hopper-hop       | 448.4 | 18.8    |
> | sac_boxd2_drop0.01 | hopper-hop       | 364.5 | 12.2    |
> | sac_boxd2_drop0.1  | hopper-hop       | 285.4 | 13.5    |
> | sac_boxd2          | hopper-stand     | 954.6 | 2.0     |
> | sac_boxd2_drop0.01 | hopper-stand     | 950.0 | 2.4     |
> | sac_boxd2_drop0.1  | hopper-stand     | 884.9 | 5.0     |
> | sac_boxd2          | humanoid-run     | 284.0 | 8.0     |
> | sac_boxd2_drop0.01 | humanoid-run     | 272.8 | 16.0    |
> | sac_boxd2_drop0.1  | humanoid-run     | 183.9 | 8.0     |
> | sac_boxd2          | humanoid-stand   | 873.2 | 5.7     |
> | sac_boxd2_drop0.01 | humanoid-stand   | 869.9 | 8.6     |
> | sac_boxd2_drop0.1  | humanoid-stand   | 769.7 | 47.9    |
> | sac_boxd2          | humanoid-walk    | 863.5 | 4.7     |
> | sac_boxd2_drop0.01 | humanoid-walk    | 854.5 | 7.8     |
> | sac_boxd2_drop0.1  | humanoid-walk    | 750.1 | 29.9    |
> | sac_boxd2          | quadruped-run    | 828.1 | 14.8    |
> | sac_boxd2_drop0.01 | quadruped-run    | 795.6 | 14.3    |
> | sac_boxd2_drop0.1  | quadruped-run    | 642.7 | 14.1    |
> | sac_boxd2          | quadruped-walk   | 951.5 | 4.1     |
> | sac_boxd2_drop0.01 | quadruped-walk   | 955.9 | 2.4     |
> | sac_boxd2_drop0.1  | quadruped-walk   | 933.9 | 7.6     |
> | sac_boxd2          | walker-run       | 813.2 | 2.0     |
> | sac_boxd2_drop0.01 | walker-run       | 771.4 | 9.7     |
> | sac_boxd2_drop0.1  | walker-run       | 667.2 | 14.6    |
>
> ---
>
> **Would mean over ensemble to do significantly worse than max over ensemble?**
>
> Thank you for the suggestion. We plan to include the experiments with mean over ensembles and we will update in an appropriate time. However, we would like to clarify again that dropout turns the NNs to an approximate Bayesian model, which enables us to take the maximum over ensembles to calculate UCB. Using the mean would simply regress to UCB with $c=0$ in $UCB(x) = \mu(x) + c\sigma(x)$.
>
>
> ---
>
> [5] Yarin Gal and Zoubin Ghahramani, “Dropout as a Bayesian Approximation: Representing Model Uncertainty in Deep Learning”, ICML 2016
> [6] Takuya Hiraoka et al., “Dropout Q-Functions for Doubly Efficient Reinforcement Learning”, ICLR 2022

---

> > ### Comment · Reviewer_jL2p · 2025-11-23
> > **Thank you for your response**
> >
> > I see, my confusion was the the non-BOXD methods are identical lines, but that was clear from the caption already -- just my mistake.
> >
> > Thank you for your reply. I still do not believe these results demonstrate that dropout or ensembles help for the reasons put forward. I don’t believe that the experiments rule out what I think is the null hypothesis, that the improvements here result from dropout regularizing (as opposed to being useful for UCB), and ensembles stabilizing (as opposed to being useful for UCB).
> >
> > Mean-of-ensemble would go some way to demonstrating that exploration is the source of improvement. Dropout=0 being worse could show that dropout is helpful, however the crux in your work is to show that it helps for exploration purposes -- something like showing that training using dropout, but not using dropout for exploration-selection, is not as good.

---

### Author Response · Authors · 2025-12-02
**Remarks by authors**

Dear reviewers and ACs,

We would like to thank you again for your valuable time in reviewing and to help improve our manuscript.

There were generally concerns regarding the novelty of our work. We would like to take this opportunity to clarify that our work’s main difference to related works is in the way to encourage optimism. We show that taking the maximum of critic ensembles, coupled with dropout, turns the NNs into an approximate Bayesian model and thus we can approximate the UCB by simply taking the maximum of these ensembles.

We show that our proposed method BOXD significantly outperforms exiting baselines (TD3, SAC, REDQ), outperforms other exploration-encouraging algorithms (OAC, DERL, TOP) and achieves on-par performance with BRO, an algorithm that employs much larger and deeper NNs, while having approximately half of BRO's training time.

Concerning reviewers requests, we have updated the manuscript colouring changes in red. We also summarise our responses to questions raised by the reviewers in the table below. We believe that our responses have addressed the reviewers’ concerns, with the resulting revisions substantially improved the manuscript.

| Topic | Raised by | Corresponding sections |
| ----     | ----     | ----  |
| Clarify theoretical justification, novelty | Reviewer n68U, Reviewer RzfA | For DERL, OAC and TOP[1], training/tuning the optimism parameter in UCB was necessary. We show that taking the maximum of critic ensembles, coupled with dropout, turns the NNs into an approximate Bayesian model and thus we can approximate the UCB by simply taking the maximum of these ensembles.  |
| Clarify metrics used | Reviewer CpNo | Updated the labels and captions of all results to clearly reflect IQM. (updated all Figures) |
| Clarify model capacity/computational cost | Reviewer CpNo, Reviewer RzfA | Revised paragraph computational cost. (added towards the end of Section 5) |
| Add dog tasks and sparse-reward tasks | Reviewer CpNo | We find our proposed method significantly outperformed baselines in both the harder dog tasks and sparse-reward manipulation tasks. (added in Section 5) |
| Add comparison to REDQ[2] |Reviewer RzfA | Our proposed method works well with REDQ and REDQ+BOXD generally outperforms REDQ significantly. (revised Figure 3 and Figure 4) |
| Add comparison to BRO[3] |Reviewer CpNo | We compare with TOP, the source of optimism of BRO, and BRO directly. Our proposed method outperforms TOP[1], and achieves best or near-best performance (outperform in some tasks) compared to BRO, while having approximately half of the computational costs. (revised Figure 3, Figure 4 too add TOP; added Appendix E. to discuss BRO; added computational costs including BRO and TOP.)  |
| Ablation on role of annealing strategy (eq. 9) | Reviewer CpNo | Perform ablation study regarding annealing strategy. Additionally, we renamed different strategies to improve clarity (revised Figure 6; revised Appendix D) |
| Ablation on role of dropout | Reviewer jL2p | Perform ablation study regarding dropout rate. (revised Appendix D) |
| Ablation on role of max vs mean | Reviewer jL2p | Perform ablation study regarding max vs mean. (revised Appendix D) |


-----

[1] T. Moskovitz et al., "Tactical Optimism and Pessimism for Deep Reinforcement Learning", NeurIPS 2021
[2] X. Chen et al., "Randomized Ensembled Double Q-Learning: Learning Fast Without a Model", ICLR 2021
[3] M Nauman et al., Bigger, "Regularized, Optimistic: scaling for compute and sample-efficient continuous control", NeurIPS 2024

---

### Meta-Review · Area_Chair_4ur9 · 2026-01-02

**Summary:**

The following are a list of reviewers' concerns:
- Limited novelty: the separate ingredients of ensembles, dropout, and disentangled exploration/exploitation policies have all been thoroughly explored before.
- Lack ablation study to identify the source of improvement. BOXD's improvement  might stem from a model capacity extension.
- Lack theoretical justification: the link between the algorithm and a principled UCB and computational principles behind Bayesian optimization is weak.
- Lack complexity analysis: The proposed method is expected to introduce significant computational overhead. The absent complexity analysis is concerning.
- The proposed “dual policy” story fails to substantiate a better exploration/exploitation balance.
- Missing citation and discussion on the related work.
- Limited experiments: lack comparison with some related work, missing experiments on sparse-reward and dog tasks.
- Missing details on metrics used.

**Reviewer Concerns:**

The authors partially addressed the following concerns:
- Limited novelty: the proposed work’s main difference to related works is in the way to encourage optimism. Taking the maximum of critic ensembles, coupled with dropout, turns the NNs into an approximate Bayesian model and thus approximate the UCB by simply taking the maximum of these ensembles.
- Missing details on metrics used: Updated the labels and captions of all results to clearly reflect IQM. (updated all Figures).
- Lack complexity analysis: Revised the paragraph on computational cost.
- Limited experiments: Added comparison to REDQ, BRO; added experiments on dog tasks and sparse-reward tasks.
- Lack ablation study: Performed the ablation study on annealing strategy, dropout, max vs mean.

The following concerns are still outstanding:
- Limited novelty
- Lack theoretical justification
- Computation cost of the proposed method
- BOXD's improvement stems from a model capacity extension
- Fail to substantiate a better exploration/exploitation balance

**Reviewer Scores:**

Reviewer jL2p raised four concerns and the authors' rebuttal did not fully address them. Reviewer jL2p might not change the score.

Reviewer n68U raised three concerns and the authors' rebuttal did not fully address them. Reviewer n68U might keep the score.

Reviewer RzfA raised four concerns and the authors' rebuttal did not fully address them. Reviewer RzfA is unlikely to change the score.

Reviewer CpNo raised four concerns and the authors' rebuttal did not fully address them. Reviewer CpNo might not change the score.

---

### Decision · Program_Chairs · 2026-01-26

Reject